# Distributionally Adaptive Meta Reinforcement Learning

**Anurag Ajay**[* †‡§], **Abhishek Gupta** [*†§], **Dibya Ghosh**[¶], **Sergey Levine**[¶], **Pulkit Agrawal**[†‡§]

Improbable AI Lab[†]
MIT-IBM Watson AI Lab[‡]
University of California, Berkeley[¶]
Massachusetts Institute Technology[§]

## Abstract

Meta-reinforcement learning algorithms provide a data-driven way to acquire policies that quickly adapt to many tasks with varying rewards or dynamics functions. However, learned meta-policies are often effective only on the exact task distribution on which they were trained and struggle in the presence of distribution shift of test-time rewards or transition dynamics. In this work, we develop a framework for meta-RL algorithms that are able to behave appropriately under test-time distribution shifts in the space of tasks. Our framework centers on an adaptive approach to distributional robustness that trains a population of meta-policies to be robust to varying levels of distribution shift. When evaluated on a potentially shifted test-time distribution of tasks, this allows us to choose the meta-policy with the most appropriate level of robustness, and use it to perform fast adaptation. We formally show how our framework allows for improved regret under distribution shift, and empirically show its efficacy on simulated robotics problems under a wide range of distribution shifts.

## 1 Introduction

The diversity and dynamism of the real world require reinforcement learning (RL) agents that can quickly adapt and learn new behaviors when placed in novel situations. Meta reinforcement learning provides a framework for conferring this ability to RL agents, by learning a "meta-policy" trained to adapt as quickly as possible to tasks from a provided training distribution [40, 11, 34, 49]. Unfortunately, meta-RL agents assume tasks to be always drawn from the training task distribution and often behave erratically when asked to adapt to tasks beyond the training distribution [5, 8]. As an example of this negative transfer, consider using meta-learning to teach a robot to navigate to goals quickly (illustrated in Figure 1). The resulting meta-policy learns to quickly adapt and walk to any target location specified in the training distribution, but explores poorly and fails to adapt to any location not in that distribution. This is particularly problematic for the meta-learning setting, since the scenarios where we need the ability to learn quickly are usually exactly those where the agent experiences distribution shift. This type of meta-distribution shift afflicts a number of real-world problems including autonomous vehicle driving [10], in-hand manipulation [18, 1, 9], and quadruped locomotion [25, 23, 19], where training task distribution may not encompass all real-world scenarios.

In this work, we study algorithms that learn meta-policies resilient to task distribution shift at test time. We assume the test-time distribution shift to be unknown but within a fixed range. One approach to enable this resiliency is leveraging the distributional robustness framework [38] to learn a meta-policy robust to a wide range of distribution shifts. However, the resulting meta-policy can be slow to adapt

---

* denotes equal contribution. Correspondence to `aajay@mit.edu` and `abhgupta@cs.washington.edu`

during test time. In contrast, learning a meta-policy robust to a small range of distribution shifts enable faster test-time adaptation but leaves the meta-policy brittle to task distribution shifts. Can we do better if we knew the degree of distribution shift apriori? Yes, but since the test-time distribution shift is unknown, we would first want our algorithm to infer the degree of distribution shift and then deploy the appropriate meta-policy robust to the inferred degree of distribution shift.

To enable our approach, we use the distributional robustness framework to train meta-policies that prepare for distribution shifts by optimizing the *worst-case* empirical risk against a set of task distributions which lie within a bounded distance from the original training task distribution (known as an *uncertainty set*). This allows meta-policies to deal with potential test-time task distribution shift, bounding their worst-case test-time regret for distributional shifts within the chosen uncertainty set.

Our key insight is that we can prepare for a variety of potential test-time distribution shifts by constructing and training against different uncertainty sets at training time. By preparing for adaptation against each of these uncertainty sets, an agent is able to adapt to a variety of potential test-time distribution shifts by adaptively choosing the most appropriate level of distributional robustness for the test distribution at hand. We introduce a conceptual framework called distributionally adaptive meta reinforcement learning, formalizing this idea. At train time, the agent learns robust meta-policies with widening uncertainty sets, preemptively accounting for different levels of test-time distribution shift that may be encountered. At test time, the agent infers the level of distribution shift it is faced with, and then uses the corresponding meta-policy to adapt to the new task (Figure 2). In doing so, the agent can adaptively choose the best level of robustness for the test-time task distribution, preserving the fast adaptation benefits of meta RL, while also ensuring good asymptotic performance under distribution shift. We instantiate a practical algorithm in this framework called DiAMetR.

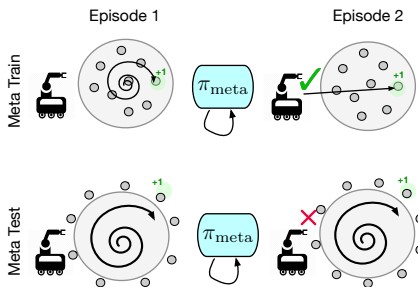

Figure 1: **Failure of Typical Meta-RL.** On meta-training tasks, $\pi_{\text{meta}}$ explores effectively and quickly learns the optimal behavior (top row). When test tasks come from a slightly larger task distribution, exploration fails catastrophically, resulting in poor adaptation behavior (bottom row).

Our main contributions are twofold. First, we show how to leverage the distributional robustness framework to make meta-reinforcement learning robust to a given level of distribution shift. Secondly, we propose a framework for making meta-reinforcement learning resilient to a variety of task distribution shifts, and DiAMetR, a practical algorithm instantiating the framework. Our experiments verify the utility of adaptive distributional robustness under test-time task distribution shift in a number of simulated robotics domains.

## 2 Related Work

Meta-reinforcement learning algorithms aim to leverage a distribution of training tasks to "learn a reinforcement learning algorithm", that is able to learn as quickly on new tasks drawn from the same distribution. A variety of algorithms have been proposed for meta-RL, including memory-based [7, 26], gradient-based [11, 36, 14] and latent-variable based [34, 49, 48, 12] schemes. These algorithms show the ability to generalize to new tasks drawn from the same distribution, and have been applied to problems ranging from robotics [28, 48, 19] to computer science education [44]. This line of work has been extended to operate in scenarios without requiring any pre-specified task distribution [13, 17], in offline settings [6, 29, 27] or in hard (meta-)exploration settings [50, 47], making them more broadly applicable to a wider class of problems. However, most meta-RL algorithms assume source and target tasks are drawn from the same distribution, an assumption rarely met in practice. Our work shows how the machinery of meta-RL can be made compatible with distribution shift at test time, using ideas from distributional robustness. Some recent work shows that model based meta-reinforcement learning can be made to be robust to a particular level distribution shift [24, 21] by learning a shared dynamics model against adversarially chosen task distributions. We show that we can build model-free meta-reinforcement learning algorithms, which are not just robust to a particular level of distribution shift, but can adapt to various levels of shift.

Distributional robustness methods have been studied in the context of building supervised learning systems that are robust to the test distribution being different than the training one. The key idea is to train a model to not just minimize empirical risk, but instead learn a model that has the

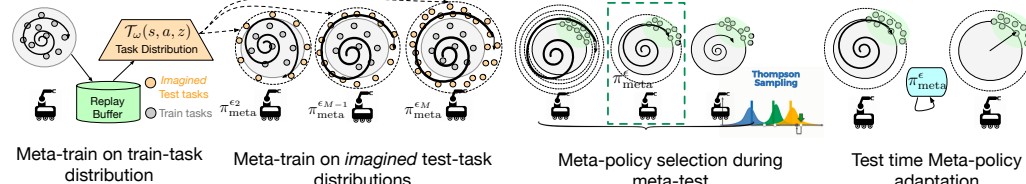

| Meta-train on train-task distribution | Meta-train on *imagined* test-task distributions | Meta-policy selection during meta-test | Test time Meta-policy adaptation |

Figure 2: During meta-train, DiAMetR learns a meta-policy $\pi_{\text{meta}}^{\epsilon_1}$ and task distribution model $\mathcal{T}_\omega(s, a, z)$ on train task distribution. Then, it uses the task distribution model to imagine different shifted test task distributions on which it learns different meta-policies $\{\pi_{\text{meta}}^{\epsilon_i}\}_{i=2}^M$, each corresponding to a different level of robustness. During meta-test, it chooses an appropriate meta-policy based on inferred test task distribution shift with Thompson's sampling and then quickly adapts the selected meta-policy to individual tasks.

lowest worst-case empirical risk among an "uncertainty-set" of distributions that are boundedly close to the empirical training distribution [38, 22, 2, 16]. If the uncertainty set and optimization are chosen carefully, these methods have been shown to obtain models that are robust to small amounts of distribution shift at test time [38, 22, 2, 16], finding applications in problems like federated learning [16] and image classification [22]. This has been extended to the min-max robustness setting for specific algorithms like model-agnostic meta-learning [3], but are critically dependent on correct specification of the appropriate uncertainty set and applicable primarily in supervised learning settings. Alternatively, several RL techniques aim to directly tackle the robustness problem, aiming to learn policies robust to adversarial perturbations [42, 46, 33, 32]. [45] conditions the policy on uncertainty sets to make it robust to different perturbation sets. While these methods are able to learn conservative, robust policies, they are unable to adapt to new tasks as DiAMetR does in the meta-reinforcement learning setting. In our work, rather than choosing a single uncertainty set, we learn many meta-policies for widening uncertainty sets, thereby accounting for different levels of test-time distribution shift.

## 3    Preliminaries

**Meta-Reinforcement Learning** aims to learn a fast reinforcement learning algorithm or a "meta-policy" that can quickly maximize performance on tasks $\mathcal{T}$ from some distribution $p(\mathcal{T})$. Formally, each task $\mathcal{T}$ is a Markov decision process (MDP) $\mathcal{M} = (\mathcal{S}, \mathcal{A}, \mathcal{P}, \mathcal{R}, \gamma, \mu_0)$; the goal is to exploit regularities in the structure of rewards and environment dynamics across tasks in $p(\mathcal{T})$ to acquire effective exploration and adaptation mechanisms that enable learning on new tasks much faster than learning the task naively from scratch. A meta-policy (or fast learning algorithm) $\pi_{\text{meta}}$ maps a history of environment experience $h \in (\mathcal{S} \times \mathcal{A} \times \mathcal{R})^*$ in a new task to an action $a$, and is trained to acquire optimal behaviors on tasks from $p(\mathcal{T})$ within $k$ episodes:

$$\min_{\pi_{\text{meta}}} \mathbb{E}_{\mathcal{T} \sim p(\mathcal{T})} \left[ \text{Regret}(\pi_{\text{meta}}, \mathcal{T}) \right],$$

$$\text{Regret}(\pi_{\text{meta}}, \mathcal{T}) = J(\pi_{\mathcal{T}}^*) - \mathbb{E}_{a_t^{(i)} \sim \pi_{\text{meta}}(\cdot | h_t^{(i)}), \mathcal{T}} \left[ \frac{1}{k} \sum_{i=1}^{k} \sum_{t=1}^{T} r_t^{(i)} \right], \quad J(\pi_{\mathcal{T}}^*) = \max_{\pi} \mathbb{E}_{\pi, \mathcal{T}} \left[ \sum_t r_t \right]$$

$$\text{where } h_t^{(i)} = (s_{1:t}^{(i)}, r_{1:t}^{(i)}, a_{1:t-1}^{(i)}) \cup (s_{1:T}^{(j)}, r_{1:T}^{(j)}, a_{1:T}^{(j)})_{j=1}^{i-1}. \tag{1}$$

Intuitively, the meta-policy has two components: an exploration mechanism that ensures that appropriate reward signal is found for all tasks in the training distribution, and an adaptation mechanism that uses the collected exploratory data to generate optimal actions for the current task. In practice, the meta-policy may be represented explicitly as an exploration policy conjoined with a policy update[11, 34], or implicitly as a black-box RNN [7, 49]. We use the terminology "meta-policies" interchangeably with that of "fast-adaptation" algorithms, since our practical implementation builds on [31] (which represents the adaptation mechanism using a black-box RNN). Our work focuses on the setting where there is potential drift between $p_{\text{train}}(\mathcal{T})$, the task distribution we have access to during training, and $p_{\text{test}}(\mathcal{T})$, the task distribution of interest during evaluation.

**Distributional robustness** [38] learns models that do not minimize empirical risk against the training distribution, but instead prepare for distribution shift by optimizing the *worst-case* empirical risk

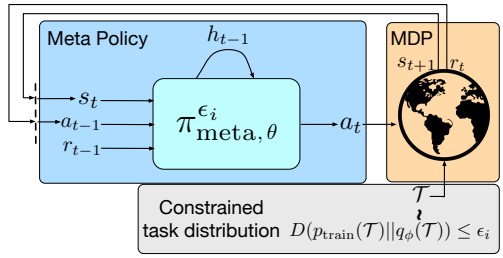
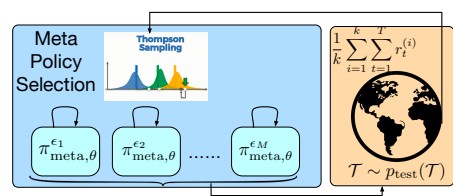

Meta Train phase

Meta Test phase

Figure 3: During meta-train phase, DiAMetR learns a family of meta-policies robust to varying levels of distribution shift (as characterized by $\epsilon_i$). During meta-test phase, given a potentially shifted test-time distribution of tasks, DiAMetR chooses the meta-policy with the most appropriate level of robustness and use it to perform fast adaptation for new tasks sampled from the same shifted test task distribution.

against a set of data distributions close to the training distribution (called an *uncertainty set*):

$$\min_{\theta} \ \max_{\phi} \mathbb{E}_{x \sim q_\phi(x)}[l(x; \theta)] \qquad \text{s.t.} \quad D(p_{\text{train}}(x)||q_\phi(x)) \leq \epsilon \qquad (2)$$

This optimization finds the model parameters $\theta$ that minimizes worst case risk $l$ over distributions $q_\phi(x)$ in an $\epsilon$-ball (measured by an $f$-divergence) from the training distribution $p_{\text{train}}(x)$.

# 4 Distributionally Adaptive Meta-Reinforcement Learning

In this section, we develop a framework for learning meta-policies, that given access to a training distribution of tasks $p_{\text{train}}(\mathcal{T})$, is still able to adapt to tasks from a test-time distribution $p_{\text{test}}(\mathcal{T})$ that is similar but not identical to the training distribution. We introduce a framework for distributionally adaptive meta-RL below and instantiate it as a practical method in Section 5.

## 4.1 Known Level of Test-Time Distribution Shift

We begin by studying a simplified problem where we can exactly quantify the degree to which the test distribution deviates from the training distribution. Suppose we know that $p_{\text{test}}$ satisfies $D(p_{\text{test}}(\mathcal{T})||p_{\text{train}}(\mathcal{T})) < \epsilon$ for some $\epsilon > 0$, where $D(\cdot||\cdot)$ is a probability divergence on the set of task distributions (e.g. an $f$-divergence [35] or a Wasserstein distance [41]). A natural learning objective to learn a meta-policy under this assumption is to minimize the *worst-case* test-time regret across any test task distribution $q(\mathcal{T})$ that is within some $\epsilon$ divergence of the train distribution:

$$\min_{\pi_{\text{meta}}} \ \mathcal{R}(\pi_{\text{meta}}, p_{\text{train}}(\mathcal{T}), \epsilon),$$

$$\mathcal{R}(\pi_{\text{meta}}, p_{\text{train}}(\mathcal{T}), \epsilon) = \max_{q(\mathcal{T})} \mathbb{E}_{\mathcal{T} \sim q(\mathcal{T})} \left[ \text{Regret}(\pi_{\text{meta}}, \mathcal{T}) \right] \qquad \text{s.t.} \quad D(p_{\text{train}}(\mathcal{T})||q(\mathcal{T})) \leq \epsilon \qquad (3)$$

Solving this optimization problem results in a meta-policy that has been trained to adapt to tasks from a *wider* task distribution than the original training distribution. It is worthwhile distinguishing this robust meta-objective, which incentivizes a *robust adaptation mechanism* to a wider set of tasks, from robust objectives in standard RL, which produce base policies robust to a wider set of dynamics conditions. The objective in Eq 3 incentivizes an agent to explore and adapt more broadly, not act more conservatively as standard robust RL methods [33] would encourage. Naturally, the quality of the robust meta-policy depends on the size of the uncertainty set. If $\epsilon$ is large, or the geometry of the divergence poorly reflect natural task variations, then the robust policy will have to adapt to an overly large set of tasks, potentially degrading the speed of adaptation.

## 4.2 Handling Arbitrary Levels of Distribution Shift

In practice, it is not known how the test distribution $p_{\text{test}}$ deviates from the training distribution, and consequently it is challenging to determine what $\epsilon$ to use in the meta-robustness objective. We propose to overcome this via an adaptive strategy: to train meta-policies for *varying* degrees of distribution shift, and at test-time, inferring which distribution shift is most appropriate through experience.

We train a population of meta-policies $\{\pi_{\text{meta}}^{(i)}\}_{i=1}^M$, each solving the distributionally robust meta-RL objective (eq 3) for a different level of robustness $\epsilon_i$:

$$\left\{\pi_{\text{meta}}^{\epsilon_i} := \arg\min_{\pi_{\text{meta}}} \mathcal{R}(\pi_{\text{meta}}, p_{\text{train}}(\mathcal{T}), \epsilon_i)\right\}_{i=1}^M \quad \text{where } \epsilon_M > \epsilon_{M-1} > \ldots > \epsilon_1 = 0 \quad (4)$$

In choosing a spectrum of $\epsilon_i$, we learn a set of meta-policies that have been trained on increasingly large set of tasks: at one end ($i = 1$), the meta-policy is trained only on the original training distribution, and at the other ($i = M$), the meta-policy trained to adapt to any possible task within the parametric family of tasks. These policies span a tradeoff between being robust to a wider set of task distributions with larger $\epsilon$ (allowing for larger distribution shifts), and being able to adapt quickly to any given task with smaller $\epsilon$ (allowing for better per-task regret minimization).

With a set of meta-policies in hand, we must now decide how to leverage test-time experience to discover the right one to use for the actual test distribution $p_{\text{test}}$. We recognize that the problem of policy selection can be treated as a stochastic multi-armed bandit problem (precise formulation in Appendix C), where pulling arm $i$ corresponds to running the meta-policy $\pi_{\text{meta}}^{\epsilon_i}$ for an entire meta-episode ($k$ task episodes). If a zero-regret bandit algorithm (eg: Thompson's sampling [43]) is used , then after a certain number of test-time meta episodes, we can guarantee that the meta-policy selection mechanism will converge to the meta-policy that best balances the tradeoff between adapting quickly while still being able to adapt to all the tasks from $p_{\text{test}}(\mathcal{T})$.

To summarize our framework for distributionally adaptive meta-RL, we train a population of meta-policies at varying levels of robustness on a distributionally robust objective that forces the learned adaptation mechanism to also be robust to tasks not in the training task distribution. At test-time, we use a bandit algorithm to select the meta-policy whose adaptation mechanism has the best tradeoff between robustness and speed of adaptation specifically on the test task distribution. Combining distributional robustness with test-time adaptation allows the adaptation mechanism to work even if distribution shift is present, while obviating the decreased performance that usually accompanies overly conservative, distributionally robust solutions.

### 4.3 Analysis

To provide some intuition on the properties of this algorithm, we formally analyze adaptive distributional robustness in a simplified meta RL problem involving tasks $\mathcal{T}_g$ corresponding to reaching some unknown goal $g$ in a deterministic MDP $\mathcal{M}$, exactly at the final timestep of an episode. We assume that all goals are reachable, and use the family of meta-policies that use a stochastic exploratory policy $\pi$ until the goal is discovered and return to the discovered goal in all future episodes. The performance of a meta-policy on a task $\mathcal{T}_g$ under this model can be expressed in terms of the state distribution of the exploratory policy: $\text{Regret}(\pi_{\text{meta}}, \mathcal{T}_g) = \frac{1}{d_\pi^T(g)}$. This particular framework has been studied in [13, 20], and is a simple, interpretable framework for analysis.

We seek to understand performance under distribution shift when the original training task distribution is relatively concentrated on a subset of possible tasks. We choose the training distribution $p_{\text{train}}(\mathcal{T}_g) = (1 - \beta)\text{Uniform}(\mathcal{S}_0) + \beta\text{Uniform}(\mathcal{S}\backslash\mathcal{S}_0)$, so that $p_{\text{train}}$ is concentrated on tasks involving a subset of the state space $\mathcal{S}_0 \subset \mathcal{S}$, with $\beta$ a parameter dictating the level of concentration, and consider test distributions that perturb under the TV metric. Our main result compares the performance of a meta-policy trained to an $\epsilon_2$-level of robustness when the true test distribution deviates by $\epsilon_1$.

**Proposition 4.1.** *Let $\overline{\epsilon_i} = \min\{\epsilon_i + \beta, 1 - \frac{|\mathcal{S}_0|}{|\mathcal{S}|}\}$. There exists $q(\mathcal{T})$ satisfying $D_{TV}(p_{train}, q) \leq \epsilon_1$ where an $\epsilon_2$-robust meta policy incurs excess regret over the optimal $\epsilon_1$-robust meta-policy:*

$$\mathbb{E}_{q(\mathcal{T})}[\text{Regret}(\pi_{meta}^{\epsilon_2}, \mathcal{T}) - \text{Regret}(\pi_{meta}^{\epsilon_1}, \mathcal{T})] \geq \left(c(\epsilon_1, \epsilon_2) + \frac{1}{c(\epsilon_1, \epsilon_2)} - 2\right) \quad (5)$$

$$\sqrt{\overline{\epsilon_1}(1 - \overline{\epsilon_1})|\mathcal{S}_0|(|\mathcal{S}| - \mathcal{S}_0|)} \quad (6)$$

*The scale of regret depends on $c(\epsilon_1, \epsilon_2) = \sqrt{\frac{\epsilon_2^{-1} - 1}{\epsilon_1^{-1} - 1}}$, a measure of the mismatch between $\epsilon_1$ and $\epsilon_2$.*

We first compare robust and non-robust solutions by analyzing the bound when $\epsilon_2 = 0$. In the regime of $\beta \ll 1$, excess regret scales as $\mathcal{O}(\epsilon_1\sqrt{\frac{1}{\beta}})$, meaning that the robust solution is most necessary

---

**Algorithm 1 DiAMetR**:Meta-training phase

1: Given: $p_{\text{train}}(\mathcal{T})$, Return: $\{\pi_{\text{meta},\theta}^{\epsilon_i}\}_{i=1}^M$
2: $\pi_{\text{meta},\theta}^{\epsilon_1}, \mathcal{D}_{\text{Replay-Buffer}} \leftarrow$ Solve equation 1 with off-policy RL$^2$
3: Prior $p_{\text{train}}(\mathcal{T}) \leftarrow$ Solve eq 8 using $\mathcal{D}_{\text{Replay-Buffer}}$
4: **for** $\epsilon$ in $\{\epsilon_2, \ldots, \epsilon_M\}$ **do**
5:     Initialize $q_\phi$, $\pi_{\text{meta},\theta}^\epsilon$ and $\lambda \geq 0$.
6:     **for** iteration $n = 1, 2, \ldots$ **do**
7:         **Meta-policy:** Update $\pi_{\text{meta},\theta}^\epsilon$ using off-policy RL$^2$ [31]

$$\theta := \theta + \alpha \nabla_\theta \mathbb{E}_{\mathcal{T} \sim q_\phi(\mathcal{T})}[\mathbb{E}_{\pi_{\text{meta},\theta}^\epsilon, \mathcal{P}_\mathcal{T}}[\frac{1}{k}\sum_{i=1}^k \sum_{t=1}^T r_\mathcal{T}(s_t^{(i)}, a_t^{(i)})]]$$

8:         **Adversarial task distribution:** Update $q_\phi$ using Reinforce [39]

$$\phi := \phi - \alpha \nabla_\phi (\mathbb{E}_{\mathcal{T} \sim q_\phi(\mathcal{T})}[\mathbb{E}_{\pi_{\text{meta},\theta}^\epsilon, \mathcal{P}_\mathcal{T}}[\frac{1}{k}\sum_{i=1}^k \sum_{t=1}^T r_\mathcal{T}(s_t^{(i)}, a_t^{(i)})]] + \lambda D_{\text{KL}}(p_{\text{train}}(\mathcal{T})\|q_\phi(\mathcal{T})))$$

9:         **Lagrange constraint multiplier:** Update $\lambda$ to enforce $D_{\text{KL}}(p_{\text{train}}\|q_\phi) < \epsilon$,

$$\lambda :=_{\lambda \geq 0} \lambda + \alpha(D_{\text{KL}}(p_{\text{train}}(\mathcal{T})\|q_\phi(\mathcal{T})) - \epsilon)$$

10:     **end for**
11: **end for**

---

when the training distribution is highly concentrated in a subset of the task space. At one extreme, if the training distribution contains no examples of tasks outside $\mathcal{S}_0$ ($\beta = 0$), the non-robust solution incurs *infinite excess regret*; at the other extreme, if the training distribution is uniform on the set of all possible tasks ($\beta = 1 - \frac{|\mathcal{S}_0|}{|\mathcal{S}|}$), *robustness provides no benefit*.

We next quantify the effect of mis-specifying the level of robustness in the meta-robustness objective, and what benefits *adaptive* distributional robustness can confer. For small $\beta$ and fixed $\epsilon_1$, the excess regret of an $\epsilon_2$-robust policy scales as $\mathcal{O}(\sqrt{\max\{\frac{\epsilon_2}{\epsilon_1}, \frac{\epsilon_1}{\epsilon_2}\}})$, meaning that excess regret gets incurred if the meta-policy is trained either to be too robust ($\epsilon_2 \gg \epsilon_1$) or not robust enough $\epsilon_1 \gg \epsilon_2$. Compared to a fixed robustness level, our strategy of training meta-policies for a sequence of robustness levels $\{\epsilon_i\}_{i=1}^M$ ensures that this misspecification constant is at most the relative spacing between robustness levels: $\max_i \frac{\epsilon_i}{\epsilon_{i-1}}$. This enables the distributionally adaptive approach to *control* the amount of excess regret by making the sequence more fine-grained, while a fixed choice of robustness incurs larger regret (as we verify empirically in our experiments as well).

## 5   DiAMetR: A Practical Algorithm for Meta-Distribution Shift

In order to instantiate our distributionally adaptive framework into a practical algorithm, we must address how task distributions should be parameterized and optimized over . We must also address how the robust meta-RL problem can be solved with stochastic gradient methods. We first introduce the individual components of task parameterization and robust optimization, describe the overall algorithm in Algorithm 1 and 2, and visualize components of DiAMetR in Fig 3.

### 5.1   Parameterizing Task Distributions

**Handling in-support distribution shifts:** For handling in-support task distribution shifts, we propose to represent new task distributions as re-weighted training task distribution $q(\mathcal{T}) \propto w_\mathcal{T} p_{\text{train}}(\mathcal{T})$ where $w_\mathcal{T} > 0$ is a parameter. Since we have a finite set of training tasks, say $\{\mathcal{T}_i\}_{i=1}^{n_{\text{tr}}}$, new task distributions become $q(\mathcal{T}_i) = \frac{w_{\mathcal{T}_i}}{\sum_{i=1}^{n_{\text{tr}}} w_{\mathcal{T}_i}}$. With a slight abuse of notation, we can write empirical training task distribution as $p_{\text{train}}(\mathcal{T}_i) = \frac{1}{n_{\text{tr}}}$. We can use KL divergence to measure the divergence between training task distribution and test task distribution $D(p_{\text{train}}(\mathcal{T})\|q_\phi(\mathcal{T})) = \sum_{i=1}^{n_{\text{tr}}} \frac{1}{n_{\text{tr}}} \log \frac{\sum_{i=1}^{n_{\text{tr}}} w_{\mathcal{T}_i}}{n_{\text{tr}} w_{\mathcal{T}_i}}$. We collectively represent the parameters $\phi = \{w_{\mathcal{T}_i}\}_{i=1}^{n_{\text{tr}}}$. Using this parameterization, the training

objective (equation 3) becomes

$$\max_{\theta} \min_{\phi} \mathbb{E}_{\mathcal{T} \sim p_{\text{train}}(\mathcal{T})} \left[ \mathbb{E}_{\pi_{\text{meta},\theta}^{\epsilon}, \mathcal{P}} \left[ \frac{n w_{\mathcal{T}}}{\sum_{i=1}^{n_{\text{tr}}} w_{\mathcal{T}_i}} \frac{1}{k} \sum_{i=1}^{k} \sum_{t=1}^{T} r_t^{(i)} \right] \right]$$
$$D_{\text{KL}}(p_{\text{train}}(\mathcal{T}) || q_{\phi}(\mathcal{T})) \le \epsilon \tag{7}$$

**Handling out-of-support distribution shift:** For handling out-of-support task distribution shifts, we propose to learn a probabilistic model of the training task distribution, and use the learned latent representation as a space on which to parameterize uncertainty sets over new task distributions. Specifically, we jointly train a task encoder $q_{\psi}(z|h)$ that encodes an environment history into the latent space, and a decoder $\mathcal{T}_{\omega}(s, a, z)$ mapping a latent vector $z$ to a property of the task using a dataset of trajectories collected from the training tasks. To describe the exact form of $\mathcal{T}_{\omega}$, we consider how tasks can differ and list two scenarios: (1) Tasks differ in reward functions: $\mathcal{T}_{\omega}$ takes form of reward functions $r_{\omega}(s, a, z)$ that maps a latent vector $z$ to a reward function and (2) Tasks differ in dynamics: $\mathcal{T}_{\omega}$ takes form of dynamics $p_{\omega}(s, a, z)$ that maps a latent vector $z$ to a dynamics function. This generative model can be trained as a standard latent variable model by maximizing a standard evidence lower bound (ELBO), trading off reward prediction and matching a prior $p_{\text{train}}(z)$ (chosen to be the unit gaussian).

$$\min_{\omega,\psi} \mathbb{E}_{h \sim \mathcal{D}} \left[ \mathbb{E}_{z \sim q_{\psi}(z|h)} \left[ \sum_{t=1}^{T} l(\mathcal{T}_{\omega}(s_t, a_t, z), h, t) \right] + D_{\text{KL}}(q_{\psi}(z|h) || \mathcal{N}(0, I)) \right]$$
$$l(\mathcal{T}_{\omega}(s_t, a_t, z), h, t) = (r_{\omega}(s_t, a_t, z) - r_t)^2 \quad \text{when rewards differ}$$
$$l(\mathcal{T}_{\omega}(s_t, a_t, z), h, t) = \|p_{\omega}(s_t, a_t, z) - s_{t+1}\|^2 \quad \text{when dynamics differ} \tag{8}$$

Having learned a latent space, we can parameterize new task distributions $q(\mathcal{T})$ as distributions $q_{\phi}(z)$ (the original training distribution corresponds to $p_{\text{train}}(z) = \mathcal{N}(0, I)$, and measure the divergence between task distributions as well using the KL divergence in this latent space $D(p_{\text{train}}(z) || q_{\phi}(z))$. Using this parameterization, the training objective (equation 3) becomes

$$\max_{\theta} \min_{\phi} \mathbb{E}_{z \sim q_{\phi}(z)} \left[ \mathbb{E}_{\pi_{\text{meta},\theta}^{\epsilon}, \mathcal{P}} \left[ \frac{1}{k} \sum_{i=1}^{k} \sum_{t=1}^{T} r_{\omega}(s_t^{(i)}, a_t^{(i)}, z) \right] \right] \quad \text{when rewards differ}$$

$$\max_{\theta} \min_{\phi} \mathbb{E}_{z \sim q_{\phi}(z)} \left[ \mathbb{E}_{\pi_{\text{meta},\theta}^{\epsilon}, p_{\omega}(\cdot,\cdot,z)} \left[ \frac{1}{k} \sum_{i=1}^{k} \sum_{t=1}^{T} r_t^{(i)} \right] \right] \quad \text{when dynamics differ}$$
$$D_{\text{KL}}(p_{\text{train}}(z) || q_{\phi}(z)) \le \epsilon \tag{9}$$

## 5.2 Training and test-time selection of meta-policies

**Learning Robust Meta-Policies:** Given this task parameterization, the next question becomes how to actually perform the robust optimization laid out in Eq:3. The distributional meta-robustness objective can be modelled as an adversarial game between a meta-policy $\pi_{\text{meta}}^{\epsilon}$ and a task proposal distribution $q(\mathcal{T})$. As described above, this task proposal distribution is parameterized as a distribution over latent space $q_{\phi}(z)$, while $\pi_{\text{meta}}^{\epsilon}$ is parameterized a typical recurrent neural network policy as in [31]. We parameterize $\{\pi_{\text{meta}}^{\epsilon_i}\}_{i=1}^{M}$ as a discrete set of meta-policies, with one for each chosen value of $\epsilon$.

This leads to a simple alternating optimization scheme (see Algorithm 1), where the meta-policy is trained using a standard meta-RL algorithm (we use off-policy RL$^2$ [31] as a base learner), and the task proposal distribution with an constrained optimization method (we use dual gradient descent [30]). Each iteration $n$, three updates are performed: 1) the meta-policy $\pi_{\text{meta},\theta}$ updated to improve performance on the current task distribution, 2) the task distribution $q_{\phi}(\mathcal{T})$ updated to increase weight on tasks where the current meta-policy adapts poorly and decreases weight on tasks that the current meta-policy can learn, while staying close to the original training distribution, and 3) a penalty coefficient $\lambda$ is updated to ensure that $q_{\phi}(\mathcal{T})$ satisfies the divergence constraint.

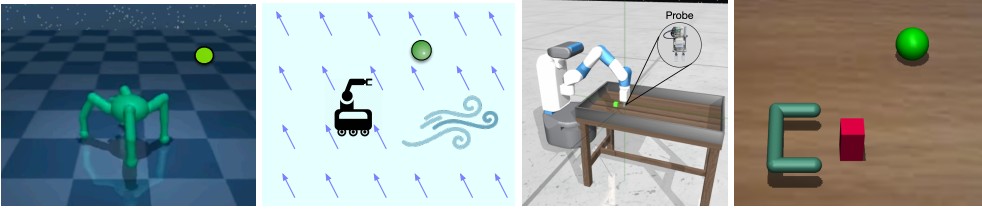

| (a) Ant navigation | (b) Wind navigation | (c) Object localization | (d) Block push |

Figure 4: The agent needs to either navigate in absence of winds, navigate in presence of winds, use its gripper to localize an object at an unobserved target location or push the block to an unobserved target location, indicated by green sphere (or cube), by exploring its environment and experiencing reward. While tasks vary in reward functions for `Ant navigation`, `Object localization` and `Block push`, they vary in dynamics function for `Wind navigation`.

---

**Algorithm 2 DiAMetR**: Meta-test phase

---

1: Given: $p_{\text{test}}(\mathcal{T}), \Pi = \{\pi_{\text{meta},\theta}^{\epsilon_i}\}_{i=1}^M$
2: Initialize `TS = Thompson-Sampler()`
3: **for** meta-episode $n = 1, 2, ...$ **do**
4:     Choose meta-policy $i$ = `TS.sample()`
5:     Run $\pi_{\text{meta},\theta}^{\epsilon_i}$ for meta-episode
6:     `TS.update(`
        `arm=i,`
        `reward=meta-episode return)`
7: **end for**

---

**Test-time meta-policy selection:** Since test-time meta-policy selection can be framed as a multi-armed bandit problem, we use a generic Thompson's sampling [43] algorithm (see Algorithm 2). Each meta-episode $n$, we sample a meta-policy $\pi_{\text{meta}}^{\epsilon}$ with probability proportional to its estimated average episodic reward, run the sampled meta-policy $\pi_{\text{meta}}^{\epsilon}$ for an meta-episode ($k$ environment episodes) and then update the estimate of the average episodic reward. Since Thompson's sampling is a zero-regret bandit algorithm, it will converge to the meta-policy that achieves the highest average episodic reward and lowest regret on the test task distribution.

## 6 Experimental Evaluation

We aim to comprehensively evaluate DiAMetR and answer the following questions: **(1)** Do meta-policies learned via DiAMetR allow for quick adaptation under different distribution shifts in the test-time task distribution? **(2)** Does learning for multiple levels of robustness actually help the algorithm adapt more effectively than a particular chosen uncertainty level? **(3)** Does proposing uncertainty sets via generative modeling provide useful distributions of tasks for robustness?

**Setup.** We use DiAMetR on four continuous control environments: `Ant navigation` (controlling a four-legged robotic quadruped), `Wind navigation` [6] (controlling a linear system robot in presence of wind), `Object localization` (controlling a Fetch robot to localize an object through its gripper) and `Block push` (controlling a robot arm to push an object) [14] (Figures 4a to 4d) (see Appendix D for details about reward function and dynamics). We design various meta RL tasks from these environments. Each meta RL task has a train task distribution $\mathcal{T}_i \sim p_{\text{train}}(\mathcal{T})$ such that each task $\mathcal{T}_i$ either parameterizes a reward function $r_i(s, a) := r(s, a, \mathcal{T}_i)$ or a dynamics function $p_i(s, a) := p(s, a, \mathcal{T}_i)$. $\mathcal{T}_i$ itself remains unobserved, the agent simply has access to reward values and executing actions in the environment. The learned meta-policies are evaluated on different distributionally shifted test task distributions $\{p_{\text{test}}^i(\mathcal{T})\}_{i=1}^K$ which are either in-support or out-of-support of training task distribution. In all meta RL tasks, the train and test task distribution is determined by the distribution of an underlying task parameter (i.e. wind velocity $w_{\mathcal{T}}$ for `Wind navigation` and target location $s_{\mathcal{T}}$ for other environments), which either determines the reward function or the dynamics function. While tasks vary in reward functions for `Ant navigation`, `Object localization` and `Block push`, they vary in dynamics function for `Wind navigation` (exact task distributions in Table 2). We use 4 random seeds for all our experiments and include the standard error bars in our plots.

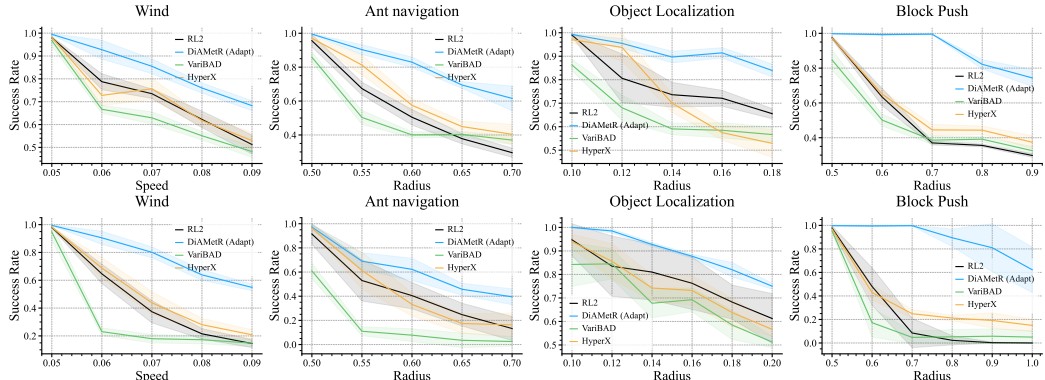

Figure 5: We evaluate DiAMetR and meta RL algorithms (RL$^2$, VariBAD and HyperX) on different *in-support* (top row) and *out-of-support* (bottom row) shifted test task distributions. DiAMetR either matches or outperforms RL$^2$, VariBAD and HyperX on these task distributions. The first point $p_1$ on the horizontal axis indicates the task parameter ($\Delta$) distribution $\mathcal{U}(0, p_1)$ and the subsequent points $p_i$ indicate task parameter ($\Delta$) distribution $\mathcal{U}(p_{i-1}, p_i)$. While task parameter for `wind navigation` is wind velocity $w_\mathcal{T}$, it is target location $s_\mathcal{T}$ for other environments. Table 2 details the task distributions used in this evaluation.

## 6.1 Adaptation to Varying Levels of Distribution Shift

During meta test, given a test task distribution $p_{\text{test}}(\mathcal{T})$, DiAMetR uses Thompson sampling to select the appropriate meta-policy $\pi_{\text{meta},\theta}^\epsilon$ within $N = 250$ meta episodes. $\pi_{\text{meta},\theta}^\epsilon$ can then solve any new task $\mathcal{T} \sim p_{\text{test}}(\mathcal{T})$ within 1 meta episode ($k$ environment episode). To test DiAMetR's ability to adapt to varying levels of distribution shift, we evaluate it on different test task distributions, as detailed in Table 2. We compare DiAMetR with meta RL algorithms such as (off-policy) RL$^2$ [31], VariBAD [49] and HyperX [50]. Since DiAMetR uses 250 meta-episodes to adaptively choose a meta-policy during test time, we finetune RL$^2$, VariBAD and HyperX with 250 meta-episodes of test task distribution to make the comparisons fair (see Appendix G for the finetuning curves). Figure 5 show that DiAMetR outperforms RL$^2$, VariBAD and HyperX on *out-of-support* and *in-support* shifted test task distributions. Furthermore, the performance gap between DiAMetR and other baselines increase as distribution shift between test task distribution and train task distribution increases. Naturally, the performance of DiAMetR also deteriorates as the distribution shift is increased, but as shown in Fig 5, it does so much more slowly than other algorithms. We also evaluate DiAMetR on train task distribution to see if it incurs any performance loss. Figure 5 shows that DiAMetR either matches or outperforms RL$^2$, VariBAD, and HyperX on the train task distribution.

## 6.2 Analysis of Tasks Proposed by Latent Conditional Uncertainty Sets

We visualize the imagined test reward distribution through their heatmaps for various distribution shifts. To generate an imagined reward function, we sample $z \sim q_\phi(z)$ and pass the $z$ into $r_\omega(s, a, z) = r_\omega(s, z)$ (given the learned reward is only dependent on state as mentioned in Appendix A.1). We then take the ant agent and reset its $(x, y)$ location to different points in the (discretized) grid $[-1, 1]^2$ and calculate $r_\omega(s, z)$ at all those points. This gives us a reward map for a single imagined reward function. We sample 10000 of these reward functions and plot them together. Figure 6 visualizes the imagined test reward distribution in `Ant-navigation` environment in increasing order of distribution shifts with respect to train reward distribution (with distribution shift parameter $\epsilon$ increasing from left to right). The train distribution of rewards has uniformly distributed target locations within the red circle. As seen in Figure 6, the learned reward distribution model imagines more target locations outside the red circle as we increase the distribution shifts.

## 6.3 Analysis of Importance of Multiple Uncertainty Sets

DiAMetR meta-learns a family of adaptation policies, each conditioned on different uncertainty set. As discussed in section 4, selecting a policy conditioned on a large uncertainty set would lead to overly conservative behavior. Furthermore, selecting a policy conditioned on a small uncertainty set would result in failure if the test time distribution shift is high. Therefore, we need to adaptively

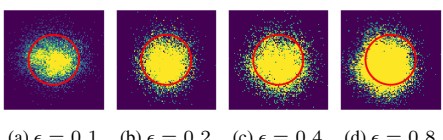

(a) $\epsilon = 0.1$    (b) $\epsilon = 0.2$    (c) $\epsilon = 0.4$    (d) $\epsilon = 0.8$

Figure 6: Imagined test reward distributions in `Ant-navigation` environment in increasing order of distribution shifts. Train reward distribution is uniform within the red circle.

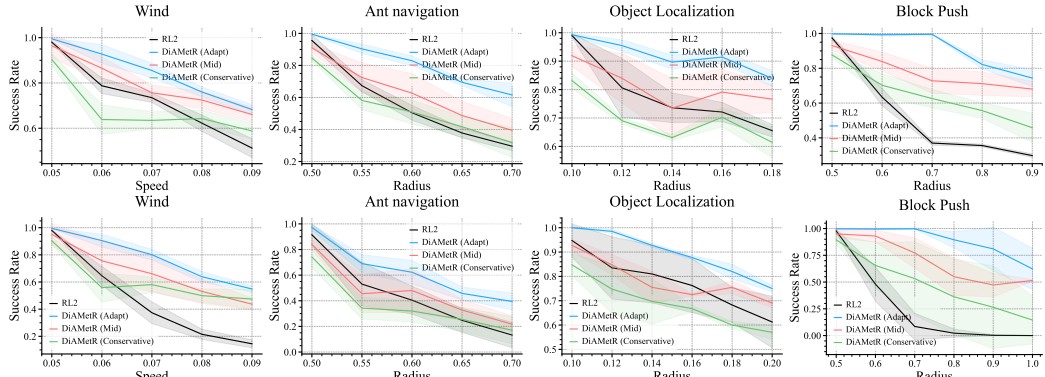

Figure 7: Adaptively choosing an uncertainty set for DiAMetR policy (Adapt) during test time allows it to better adapt to test time distribution shift than choosing an uncertainty set beforehand (Mid). Choosing a large uncertainty set for DiAMetR policy (Conservative) leads to a conservative behavior and hurts its performance when test time distribution shift is low. While top row contains *in-support* task distribution shifts, the bottom row contains *out-of-support* task distribution shifts.

select an uncertainty set during test time. To validate this phenomenon empirically, we performed an ablation study in Figure 7. As clearly visible, adaptively choosing an uncertainty set during test time allows for better test time distribution adaptation when compared to selecting an uncertainty set beforehand or selecting a large uncertainty set. These results suggest that a combination of training robust meta-learners and constructing various uncertainty sets allows for effective test-time adaptation under distribution shift. DiAMetR is able to avoid both overly conservative behavior and under-exploration at test-time.

# 7 Discussion

In this work, we introduce a distributionally "adaptive" meta RL framework for tackling task distribution shifts and argue for adaptation in face of these distribution shifts. There are several avenues for future work we are keen on exploring, for instance extending adaptive distributional robustness to more complex meta RL tasks, including those with image observations. Another interesting direction would be to develop a more formal theory providing adaptive robustness guarantees in meta-RL problems under these inherent distribution shifts.

## Acknowledgements

The authors thank the members of Improbable AI Lab & RAIL for discussions and helpful feedback. We thank MIT Supercloud and the Lincoln Laboratory Supercomputing Center for providing compute resources. This research was supported by an NSF graduate fellowship, a DARPA Machine Common Sense grant, a MURI grant from the Army Research Office under the Cooperative Agreement Number W911NF-21-1-0097, and an MIT-IBM grant. This research was also partly sponsored by the United States Air Force Research Laboratory and the United States Air Force Artificial Intelligence Accelerator and was accomplished under Cooperative Agreement Number FA8750-19-2-1000. The views and conclusions contained in this document are those of the authors and should not be interpreted as representing the official policies, either expressed or implied, of the United States Air Force, the United States Army Research Office or the U.S. Government. The U.S. Government is authorized to reproduce and distribute reprints for Government purposes, notwithstanding any copyright notation herein.

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
