# A DiAMetR for handling out-of-support task distribution shifts

To handle *out-of-support* task distribution shifts, we parameterized new task distribution as latent space distributions $q_\phi(z)$ and measure the divergence between train and test task distribution via KL divergence in latent space $D(p_{\text{train}}(z)||q_\phi(z))$ in Section 5. Using this parameterization, equation 3 becomes

$$\max_\theta \min_\phi \mathbb{E}_{z \sim q_\phi(z)} \left[ \mathbb{E}_{\pi^\epsilon_{\text{meta},\theta}, \mathcal{P}} \left[ \frac{1}{k} \sum_{i=1}^k \sum_{t=1}^T r_\omega(s_t^{(i)}, a_t^{(i)}, z) \right] \right] \quad \text{when rewards differ}$$

$$\max_\theta \min_\phi \mathbb{E}_{z \sim q_\phi(z)} \left[ \mathbb{E}_{\pi^\epsilon_{\text{meta},\theta}, p_\omega(\cdot,\cdot,z)} \left[ \frac{1}{k} \sum_{i=1}^k \sum_{t=1}^T r_t^{(i)} \right] \right] \quad \text{when dynamics differ}$$

$$D_{\text{KL}}(p_{\text{train}}(z)||q_\phi(z)) \leq \epsilon \tag{10}$$

This objective function is solved in Algorithm 1 for different values of $\epsilon$. To imagine *out-of-support* distributionally shifted task (i.e. reward or dynamics) distributions, DiAMetR leverages structured VAE which we describe in subsequent subsections.

## A.1 Structured VAE for modeling reward distributions

We leverage the sparsity in reward functions (i.e. $0/1$ rewards) in the environments used and describe a structured VAE to model $r_\omega(s, a, z)$ with $p(z) = \mathcal{N}(0, I)$ and KL-divergence for $D(\cdot||\cdot)$. Let $\overline{h} = (\sum_{t=1}^T r_t s_t)/(\sum_{t=1}^T r_t)$ be the mean of states achieving a $+1$ reward in trajectory $h$. The encoder $z \sim q_\psi(z|\overline{h})$ encodes $\overline{h}$ into a latent vector $z$. The reward model $r_\omega(s, a, z)$ consists of 2 components: (i) latent decoder $\hat{\overline{h}} = r_\omega^h(z)$ which reconstructs $\overline{h}$ and (ii) reward predictor $r_\omega^{\text{rew}}(s, \hat{\overline{h}}) = \exp(-\|M \odot (s - \hat{\overline{h}})\|_2^2/\sigma^2)$ which predicts reward for a state given the decoded latent vector. $M$ is a masking function and $\sigma$ is a learned parameter. The training objective becomes

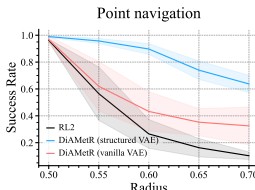

Point navigation

Figure 8: Using a vanilla VAE, in lieu of a structured VAE, to model task distribution hurts DiAMetR's performance on test-task distributions.

$$\min_{\omega,\psi} \mathbb{E}_{h \sim \mathcal{D}} \left[ \mathbb{E}_{z \sim q_\psi(z|\overline{h})} \left[ \left\| \overline{h} - r_\omega^h(z) \right\|_2^2 + \sum_{t=1}^T \left\| r_\omega^{\text{rew}}(s_t, r_\omega^h(z)) - r_t \right\|^2 \right] + D_{\text{KL}}(q_\psi(z|\overline{h})||p(z)) \right] \tag{11}$$

The structure in the VAE helps in extrapolating reward functions when $z \sim q_\phi(z)$. This can be further verified by reduction in DiAMetR's performance on test-task distributions when using vanilla VAE (see Figure 8).

## A.2 Structured VAE for modelling dynamics distributions

We describe our structured VAE architecture for modelling dynamics distribution $p_\omega(s, a, z)$ with $p(z) = \mathcal{N}(0, I)$ and KL-divergence for $D(\cdot||\cdot)$. It handles *out-of-support* shifted test task distributions where only dynamics vary across tasks. We leverage the fact that dynamics differ by an additive term. The encoder $z \sim q_\phi(z|(s_t, a_t)_{t=1}^T)$ encodes the state action trajectory to a latent vector $z$. The dynamics model takes the form $p_\omega(s, a, z) = W_\omega z + p_\omega^{\text{dyn}}(s, a)$ where $W_\omega$ is a parameter. The training objective becomes:

$$\min_{\omega,\psi} \mathbb{E}_{(s_t,a_t)_{t=1}^T \sim \mathcal{D}} \left[ \mathbb{E}_{z \sim q_\psi(z|(s_t,a_t)_{t=1}^T)} \left[ \sum_{t=1}^{T-1} \left\| p_\omega^{\text{dyn}}(s_t, a_t) + W_\omega z - s_{t+1} \right\|^2 \right] + D_{\text{KL}}(q_\psi(z|(s_t, a_t)_{t=1}^T)||p(z)) \right] \tag{12}$$

The structure in the VAE helps in extrapolating dynamics when $z \sim q_\phi(z)$.

**Algorithm 3** (Detailed) **DiAMetR**:Meta-training phase

1: Given: $p_{\text{train}}(\mathcal{T})$, Return: $\{\pi_{\text{meta},\theta}^{\epsilon_i}\}_{i=1}^M$
2: $\pi_{\text{meta},\theta}^{\epsilon_1}, \mathcal{D}_{\text{Replay-Buffer}} \leftarrow$ Solve equation 1 with off-policy RL[2]
3: Reward distribution $r_\omega$ / Dynamic distribution $p_\omega$, prior $p_{\text{train}}(z) \leftarrow$ Solve eq 8 using $\mathcal{D}_{\text{Replay-Buffer}}$
4: **for** $\epsilon$ in $\{\epsilon_2, \ldots, \epsilon_M\}$ **do**
5:    Initialize $q_\phi$, $\pi_{\text{meta},\theta}^{\epsilon}$ and $\lambda \geq 0$.
6:    **for** iteration $n = 1, 2, \ldots$ **do**
7:       **Meta-policy:** Update $\pi_{\text{meta},\theta}^{\epsilon}$ using off-policy RL[2] [31]

$$\theta := \theta + \alpha \nabla_\theta \mathbb{E}_{\mathcal{T} \sim p_{\text{train}}(\mathcal{T})} [\mathbb{E}_{\pi_{\text{meta},\theta}^{\epsilon}, \mathcal{T}} [\frac{n w_{\mathcal{T}}}{k \sum_{i=1}^{n_{\text{tr}}} w_{\mathcal{T}_i}} \sum_{i=1}^{k} \sum_{t=1}^{T} r_t^{(i)}]]$$

$$\theta := \theta + \alpha \nabla_\theta \mathbb{E}_{z \sim q_\phi(z)} [\mathbb{E}_{\pi_{\text{meta},\theta}^{\epsilon}, \mathcal{P}} [\frac{1}{k} \sum_{i=1}^{k} \sum_{t=1}^{T} r_\omega(s_t^{(i)}, a_t^{(i)}, z)]]$$

$$\theta := \theta + \alpha \nabla_\theta \mathbb{E}_{z \sim q_\phi(z)} [\mathbb{E}_{\pi_{\text{meta},\theta}^{\epsilon}, p_\omega(\cdot, \cdot, z)} [\frac{1}{k} \sum_{i=1}^{k} \sum_{t=1}^{T} r_t^{(i)}]]$$

8:       **Adversarial task distribution:** Update $q_\phi$ using Reinforce [39]

$$\phi := \phi - \alpha \nabla_\phi (\mathbb{E}_{\mathcal{T} \sim p_{\text{train}}(\mathcal{T})} [\mathbb{E}_{\pi_{\text{meta},\theta}^{\epsilon}, \mathcal{T}} [\frac{n w_{\mathcal{T}}}{k \sum_{i=1}^{n_{\text{tr}}} w_{\mathcal{T}_i}} \sum_{i=1}^{k} \sum_{t=1}^{T} r_t^{(i)}]] + \lambda D_{\text{KL}}(p_{\text{train}}(\mathcal{T}) \| q_\phi(\mathcal{T}))$$

$$\phi := \phi - \alpha \nabla_\phi (\mathbb{E}_{z \sim q_\phi(z)} [\mathbb{E}_{\pi_{\text{meta},\theta}^{\epsilon}, \mathcal{P}} [\frac{1}{k} \sum_{i=1}^{k} \sum_{t=1}^{T} r_\omega(s_t^{(i)}, a_t^{(i)}, z)]] + \lambda D_{\text{KL}}(p_{\text{train}}(z) \| q_\phi(z))$$

$$\phi := \phi - \alpha \nabla_\phi (\mathbb{E}_{z \sim q_\phi(z)} [\mathbb{E}_{\pi_{\text{meta},\theta}^{\epsilon}, p_\omega(\cdot, \cdot, z)} [\frac{1}{k} \sum_{i=1}^{k} \sum_{t=1}^{T} r_t^{(i)}]] + \lambda D_{\text{KL}}(p_{\text{train}}(z) \| q_\phi(z))$$

9:       **Lagrange constraint multiplier:** Update $\lambda$ to enforce $D_{\text{KL}}(p_{\text{train}} \| q_\phi) < \epsilon$,

$$\lambda :=_{\lambda \geq 0} \lambda + \alpha (D_{\text{KL}}(p_{\text{train}}(\mathcal{T}) \| q_\phi(\mathcal{T})) - \epsilon)$$
$$\lambda :=_{\lambda \geq 0} \lambda + \alpha (D_{\text{KL}}(p_{\text{train}}(z) \| q_\phi(z)) - \epsilon) \quad \lambda :=_{\lambda \geq 0} \lambda + \alpha (D_{\text{KL}}(p_{\text{train}}(z) \| q_\phi(z)) - \epsilon)$$

10:   **end for**
11: **end for**

# B  DiAMetR for handling in-support task distribution shifts

To handle *in-support* task distribution shifts, we parameterized new task distribution as re-weighted (empirical) training task distribution $q_\phi(\mathcal{T}) \propto w_{\mathcal{T}} p_{\text{train}}(\mathcal{T})$ (where $\phi = \{w_{\mathcal{T}_i}\}_{i=1}^{n_{\text{tr}}}$) in Section 5. Using this parameterization, equation 3 becomes

$$\max_\theta \min_\phi \mathbb{E}_{\mathcal{T} \sim p_{\text{train}}(\mathcal{T})} \left[ \mathbb{E}_{\pi_{\text{meta},\theta}^{\epsilon}, \mathcal{P}} \left[ \frac{n w_{\mathcal{T}}}{\sum_{i=1}^{n_{\text{tr}}} w_{\mathcal{T}_i}} \frac{1}{k} \sum_{i=1}^{k} \sum_{t=1}^{T} r_t^{(i)} \right] \right]$$
$$D_{\text{KL}}(p_{\text{train}}(\mathcal{T}) \| q_\phi(\mathcal{T})) \leq \epsilon \tag{13}$$

This objective function is solved in Algorithm 1 for different values of $\epsilon$. For *in-support* task distribution shifts, shifts in dynamics distribution and reward distribution don't require separate treatment.

# C  Test time Meta Policy Selection

As discussed in Section 4, to adapt to test time task distribution shifts, we train a family of meta-policies $\Pi = \{\pi_{\text{meta}}^{\epsilon_i}\}$ to be robust to varying degrees of distribution shifts. We then choose the appropriate meta-policy during test-time based on the inferred task distribution shift. In this section, we frame the test-time selection of meta-policy from the family $\Pi$ as a stochastic multi-arm bandit problem. Every iteration involves pulling an arm $i$ which corresponds to executing $\pi_{\text{meta}}^{\epsilon_i}$ for 1 meta-episode ($k$ environment episodes) on a task $\mathcal{T} \sim p_{\text{test}}(\mathcal{T})$. Let $R_i$ be the expected return for pulling

arm $i$

$$R_i = \mathbb{E}_{\pi_{\text{meta}}^{\epsilon_i}, \mathcal{T} \sim p_{\text{test}}(\mathcal{T})} \left[ \frac{1}{k} \sum_{i=1}^{k} \sum_{t=1}^{T} r_t^{(i)} \right] \tag{14}$$

Let $R^* = \max_{i \in \{1,\ldots,M\}} R_i$ and $\pi_{\text{meta}}^{\epsilon}$ be the corresponding meta-policy. The goal of the stochastic bandit problem is to pull arms $i_1, \ldots, i_N \in \{1, \ldots, M\}$ such that the test-time regret $\mathcal{R}_N$ is minimized

$$\mathcal{R}_N^{\text{test}} = N R^* - \sum_{t=1}^{N} R_{i_t} \tag{15}$$

with constraint that $i_t$ can depend only on the information available prior to iteration $t$. We choose Thompson sampling, a zero-regret bandit algorithm, to solve this problem. In principle, Thompson sampling should learn to choose $\pi_{\text{meta}}^{\epsilon}$ after $N$ iterations.

## D    Environment Description

We describe the environments used in the paper:

- `{Point,Wheeled,Ant}-navigation`: The rewards for each task correspond to reaching an unobserved target location $s_{\mathcal{T}}$. The agent (i.e. Wheeled, Ant) must explore the environment to find the unobserved target location (Wheeled driving a differential drive robot, Ant controlling a four legged robotic quadruped). It receives a reward of $1$ once it gets within a small $\delta$ distance of the target $s_t$, as in [14].

- `Dense Ant-navigation`: The rewards for each task correspond to reaching an unobserved target location $s_{\mathcal{T}}$. The Ant (a four legged robotic quadruped) must explore the environment to find the unobserved target location. It receives a reward of $-\|\text{agent}_{x,y} - s_{\mathcal{T}}\|$ where $\text{agent}_{x,y}$ is the (x,y) position of the Ant.

- `Wind navigation`: The rewards for each task correspond to reaching an unobserved target location $s_t$. While $s_{\mathcal{T}}$ is fixed across tasks (i.e. at $(1/\sqrt{2}, 1/\sqrt{2})$), the agent (a linear system robot) much navigate in the presence of wind (i.e. a noise vector $w_{\mathcal{T}}$) that varies across tasks. It receives a reward of $1$ once it gets within a small $\delta$ distance of the target $s_{\mathcal{T}}$.

- `Object localization`: Each task corresponds to using the gripper to localize an object kept at an unobserved target location $s_t$. The Fetch robot must move its gripper around and explore the environment to find the object. Once the gripper touches the object kept at target location $s_t$, it receives a reward of $1$.

- `Block push`: Each task corresponds to moving the block to an unobserved target location $s_t$. The robot arm must move the block around and explore the environment to find the unobserved target location. Once the block gets within a small $\delta$ distance of the target $s_t$, it receives a reward of $1$, as in [14].

Furthermore, Table 1 describes the state space $\mathcal{S}$, action space $\mathcal{A}$, episodic horizon $H$, frameskip for each environment and $k$ (i.e. number of environment episodes in 1 meta episode). Table 2 provides parameters for train and test task distributions for different meta-RL tasks used in the paper.

| Name | State space $\mathcal{S}$ | Action space $\mathcal{A}$ | Episodic Horizon $H$ | Frameskip | $k$ |
|---|---|---|---|---|---|
| Point-navigation | Box(2,) | Box(2,) | 60 | 1 | 2 |
| Wind-navigation | Box(2,) | Box(2,) | 25 | 1 | 1 |
| Wheeled-navigation | Box(12,) | Box(2,) | 60 | 10 | 2 |
| Ant-navigation | Box(29,) | Box(8,) | 200 | 5 | 2 |
| Dense Ant-navigation | Box(29,) | Box(8,) | 200 | 5 | 2 |
| Object localization | Box(17,) | Box(6,) | 50 | 10 | 2 |
| Block Push | Box(10,) | Box(4,) | 60 | 10 | 2 |

Table 1: Environment Description

| Environment | Task type | Task parameter distribution | $p_{\text{train}}(\mathcal{T})$ | $\{p^i_{\text{test}}(\mathcal{T})\}^K_{i=1}$ |
|---|---|---|---|---|
| Point, Wheeled, Ant-navigation | reward change, out-of-support shift | $s_{\mathcal{T}} = (\Delta\cos\theta, \Delta\sin\theta)$
$\Delta \sim \mathcal{U}(\Delta_{\min}, \Delta_{\max})$
$\theta \sim \mathcal{U}(0, 2\pi)$ | $\Delta \sim \mathcal{U}(0, 0.5)$ | $\Delta \sim \mathcal{U}(0, 0.5), \mathcal{U}(0.5, 0.55)$
$\mathcal{U}(0.55, 0.6), \mathcal{U}(0.6, 0.65)$
$\mathcal{U}(0.65, 0.7)$ |
| Point, Wheeled, Ant-navigation | reward change, in-support shift | $s_{\mathcal{T}} = (\Delta\cos\theta, \Delta\sin\theta)$
$\Delta \sim \mathcal{U}(\Delta_{\min}, \Delta_{\max})$
$\theta \sim \mathcal{U}(0, 2\pi)$ | $\Delta \sim \text{Exp}(\lambda = 5)$ | $\Delta \sim \mathcal{U}(0, 0.5), \mathcal{U}(0.5, 0.55)$
$\mathcal{U}(0.55, 0.6), \mathcal{U}(0.6, 0.65)$
$\mathcal{U}(0.65, 0.7)$ |
| Dense Ant-navigation | reward change, in-support shift | $s_{\mathcal{T}} = (\Delta\cos\theta, \Delta\sin\theta)$
$\Delta \sim \mathcal{U}(\Delta_{\min}, \Delta_{\max})$
$\theta \sim \mathcal{U}(0, 2\pi)$ | $\Delta \sim \text{Exp}(\lambda = 5)$ | $\Delta \sim \mathcal{U}(0, 0.5), \mathcal{U}(0.5, 0.55)$
$\mathcal{U}(0.55, 0.6), \mathcal{U}(0.6, 0.65)$
$\mathcal{U}(0.65, 0.7)$ |
| Wind -navigation | dynamics change, out-of-support shift | $w_{\mathcal{T}} = (\Delta\cos\theta, \Delta\sin\theta)$
$\Delta \sim \mathcal{U}(\Delta_{\min}, \Delta_{\max})$
$\theta \sim \mathcal{U}(0, 2\pi)$ | $\Delta \sim \mathcal{U}(0, 0.05)$ | $\Delta \sim \mathcal{U}(0, 0.05), \mathcal{U}(0.05, 0.06)$
$\mathcal{U}(0.06, 0.07), \mathcal{U}(0.07, 0.08)$
$\mathcal{U}(0.08, 0.09)$ |
| Wind -navigation | dynamics change, in-support shift | $w_{\mathcal{T}} = (\Delta\cos\theta, \Delta\sin\theta)$
$\Delta \sim \mathcal{U}(\Delta_{\min}, \Delta_{\max})$
$\theta \sim \mathcal{U}(0, 2\pi)$ | $\Delta \sim \text{Exp}(\lambda = 40)$ | $\Delta \sim \mathcal{U}(0, 0.05), \mathcal{U}(0.05, 0.06)$
$\mathcal{U}(0.06, 0.07), \mathcal{U}(0.07, 0.08)$
$\mathcal{U}(0.08, 0.09)$ |
| Object localization | reward change, out-of-support shift | $s_{\mathcal{T}} = (\Delta\cos\theta, \Delta\sin\theta)$
$\Delta \sim \mathcal{U}(\Delta_{\min}, \Delta_{\max})$
$\theta \sim \mathcal{U}(0, 2\pi)$ | $\Delta \sim \mathcal{U}(0, 0.1)$ | $\Delta \sim \mathcal{U}(0, 0.1), \mathcal{U}(0.1, 0.12)$
$\mathcal{U}(0.12, 0.14), \mathcal{U}(0.14, 0.16)$
$\mathcal{U}(0.16, 0.18), \mathcal{U}(0.18, 0.2)$ |
| Object localization | reward change, in-support shift | $s_{\mathcal{T}} = (\Delta\cos\theta, \Delta\sin\theta)$
$\Delta \sim \mathcal{U}(\Delta_{\min}, \Delta_{\max})$
$\theta \sim \mathcal{U}(0, 2\pi)$ | $\Delta \sim \text{Exp}(\lambda = 20)$ | $\Delta \sim \mathcal{U}(0, 0.1), \mathcal{U}(0.1, 0.12)$
$\mathcal{U}(0.12, 0.14), \mathcal{U}(0.14, 0.16)$
$\mathcal{U}(0.16, 0.18)$ |
| Block-push | reward change, out-of-support shift | $s_{\mathcal{T}} = (\Delta\cos\theta, \Delta\sin\theta)$
$\Delta \sim \mathcal{U}(\Delta_{\min}, \Delta_{\max})$
$\theta \sim \mathcal{U}(0, \pi/2)$ | $\Delta \sim \mathcal{U}(0, 0.5)$ | $\Delta \sim \mathcal{U}(0, 0.5), \mathcal{U}(0.5, 0.6)$
$\mathcal{U}(0.6, 0.7), \mathcal{U}(0.7, 0.8)$
$\mathcal{U}(0.8, 0.9), \mathcal{U}(0.9, 1.0)$ |
| Block-push | reward change, in-support shift | $s_{\mathcal{T}} = (\Delta\cos\theta, \Delta\sin\theta)$
$\Delta \sim \mathcal{U}(\Delta_{\min}, \Delta_{\max})$
$\theta \sim \mathcal{U}(0, \pi/2)$ | $\Delta \sim \text{Exp}(\lambda = 4)$ | $\Delta \sim \mathcal{U}(0, 0.5), \mathcal{U}(0.5, 0.6)$
$\mathcal{U}(0.6, 0.7), \mathcal{U}(0.7, 0.8)$
$\mathcal{U}(0.8, 0.9), \mathcal{U}(0.9, 1.0)$ |

Table 2: Parameters for train and test task distribution for {Point,Wheeled, Ant}-navigation, Dense Ant-navigation, Wind-navigation, Object localization and Block-push. While tasks in Wind-navigation vary in dynamics, tasks in other environments vary in reward function. The shifted test task distributions can be either *in-support* or *out-of-support* of the training task distribution. All these task distributions are determined by distributions of underlying task parameters (say target location $s_{\mathcal{T}}$ or wind velocity $w_{\mathcal{T}}$), which either determine the reward function or the dynamics function.

# E   Experimental Evaluation on Wheeled and Point Robot Navigation

In Section 6, we evaluated DiAMetR on Wind-navigation, Ant-navigation, Fetch-reach and Block-push. We continue the experimental evaluation of DiAMetR in this section and compare it to $\text{RL}^2$, VariBAD, and HyperX on train task distribution and different test task distributions of Point navigation and Wheeled navigation [14]. We see that DiAMetR either matches or outperforms the baselines on train task distribution and outperforms the baselines on test task distributions. Furthermore, adaptively selecting an uncertainty set during test time allows for better test time distribution adaptation when compared to selecting an uncertainty set beforehand or selecting a large uncertainty set.

# F   Evaluations on dense reward environments with in-support distribution shifts

We test the applicability of DiAMetR on an environment with dense rewards. We use a variant of Ant navigation, namely Dense Ant-navigation for this evaluation. Furthermore, the shifted test task distributions are *in-support* of the training task distribution (see Table 2 for a detailed description of these task distributions). Figure 10 shows that DiAMetR still outperforms existing meta RL algorithms ($\text{RL}^2$, VariBAD, HyperX) on shifted test task distributions. However, the gap between DiAMetR and other meta RL algorithms is less than in sparse reward environments.

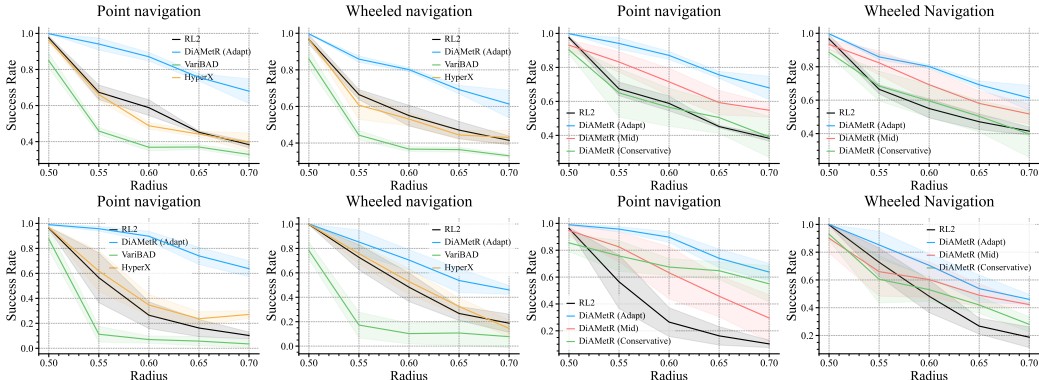

Figure 9: We evaluate DiAMetR and meta RL algorithms ($RL^2$, VariBAD and HyperX) on different *in-support* and *out-of-support* shifted test task distribution of `Point navigation` and `Wheeled navigation`. DiAMetR either matches or outperforms $RL^2$, VariBAD and HyperX on these task distributions. Furthermore, adaptively selecting an uncertainty set of DiAMetR policy (Adapt) during test time allows it to better adapt to test time distribution shift than choosing an uncertainty set beforehand (Mid). Choosing a large uncertainty set of DiAMetR policy (Conservative) leads to a conservative behavior and hurts its performance when test time distribution shift is low. The first point $p_1$ on the horizontal axis indicates the task parameter ($\Delta$) distribution $\mathcal{U}(0, p_1)$ and the subsequent points $p_i$ indicate task parameter ($\Delta$) distribution $\mathcal{U}(p_{i-1}, p_i)$. Here, task parameter is target location $s_\mathcal{T}$. Table 2 details the task distributions used in this evaluation.

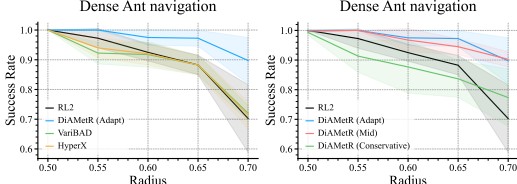

Figure 10: We evaluate DiAMetR and meta RL algorithms ($RL^2$, VariBAD and HyperX) on different *in-support* shifted test task distributions of `Dense Ant-navigation`. DiAMetR either matches or outperforms $RL^2$, VariBAD and HyperX on these test task distributions. Furthermore, selecting a single uncertainty set (that is neither too small nor too large) is sufficient in `Dense Ant-navigation` as DiAMetR(Adapt) and DiAMetR(Mid) have similar performances (within standard error). The first point $p_1$ on the horizontal axis indicates the task target distance ($\Delta$) distribution $\mathcal{U}(0, p_1)$ and the subsequent points $p_i$ indicate task target distance ($\Delta$) distribution $\mathcal{U}(p_{i-1}, p_i)$.

Furthermore, Figure 10 shows that selecting a single uncertainty set (that is neither too small nor too large) is sufficient in `Dense Ant-navigation` as DiAMetR(Adapt) and DiAMetR(Mid) have similar performances (within standard error).

# G   Meta-policy Selection and Adaptation during Meta-test

In this section, we show that DiAMetR is able adapt to various test task distributions across different environments by selecting an appropriate meta-policy based on the inferred test-time distribution shift and then quickly adapting the meta-policy to new tasks drawn from the same test-distribution. The performance of meta-RL baselines ($RL^2$, variBAD, HyperX) remains more or less the same after test-time finetuning showing that 10 iteration (with 25 rollouts per iteration) isn't enough for the meta-RL baselines to adapt to a new task distribution. For comparison, these meta-RL baselines take 1500 iterations (with 25 meta-episodes per iteration) during training to learn a meta-policy for train task distribution.

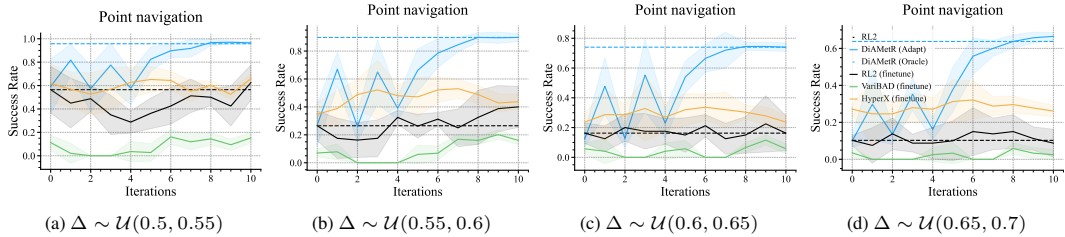

Figure 11: We compare test time adaptation of DiAMetR with test time finetuning of RL$^2$ on point robot navigation for various test task distributions. We run the adaptation procedure for 10 iterations collecting 25 rollouts per iteration.

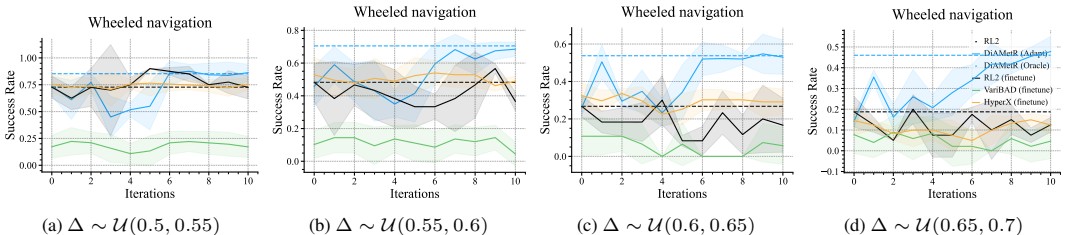

Figure 12: We compare test time adaptation of DiAMetR with test time finetuning of RL$^2$ on wheeled navigation for various test task distributions. We run the adaptation procedure for 10 iterations collecting 25 rollouts per iteration.

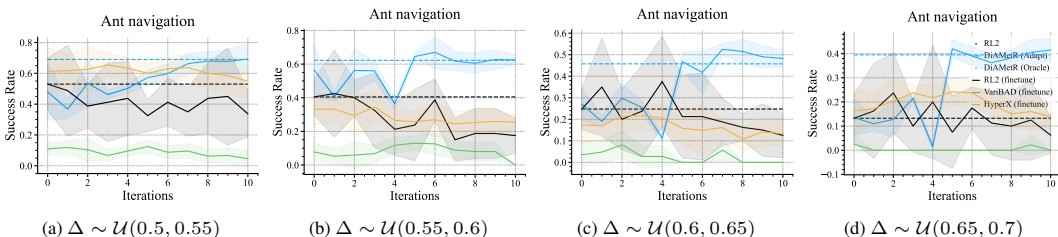

Figure 13: We compare test time adaptation of DiAMetR with test time finetuning of RL$^2$ on ant navigation for various test task distributions. We run the adaptation procedure for 10 iterations collecting 25 rollouts per iteration.

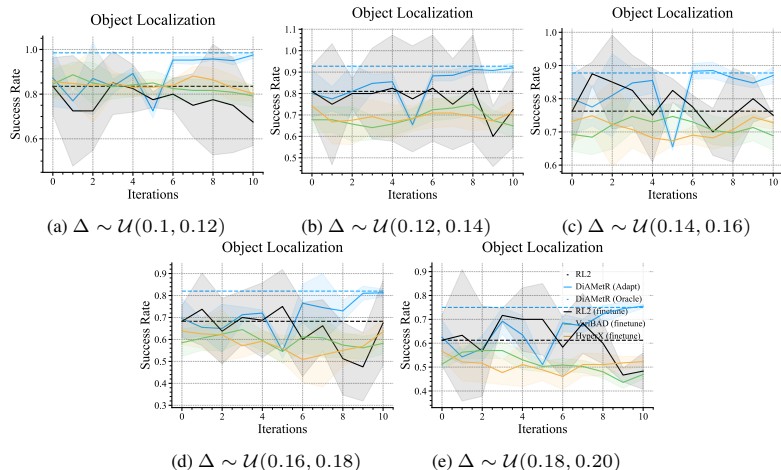

Figure 14: We compare test time adaptation of DiAMetR with test time finetuning of RL$^2$ on object localization for various test task distributions. We run the adaptation procedure for 10 iterations collecting 25 rollouts per iteration.

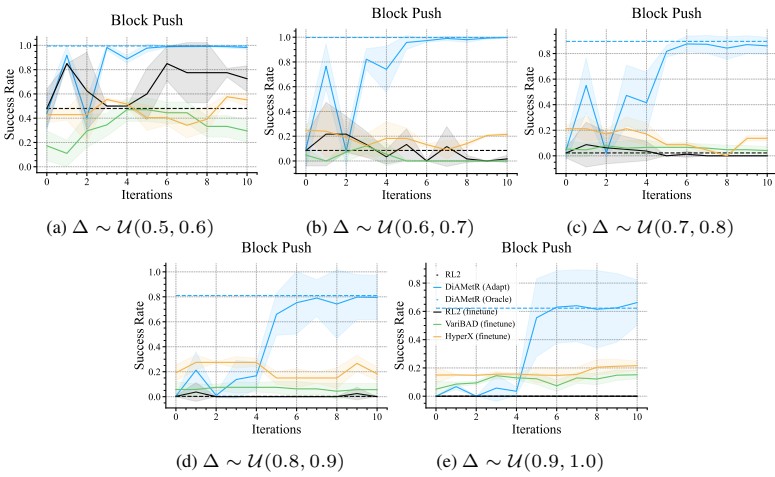

Figure 15: We compare test time adaptation of DiAMetR with test time finetuning of RL$^2$ on block push for various test task distributions. We run the adaptation procedure for 10 iterations collecting 25 rollouts per iteration.

## H Ablation studies

**Can meta RL achieve robustness to task distribution shifts with improved meta-exploration?**
To test if improved meta-exploration can help meta-RL algorithm achieve robustness to test-time task distribution shifts, we test HyperX [50] on test-task distributions in different environments. HyperX leverages curiosity-driven exploration to visit novel states for improved meta-exploration during meta-training. Despite improved meta-exploration, HyperX fails to adapt to test-time task distribution shifts (see Figure 5 and Figure 9 for results on results on different environments). This is because HyperX aims to minimize regret on train-task distribution and doesn't leverage the visited novel states to learn new behaviors helpful in adapting to test-time task distribution shifts. Furthermore, we note that the contributions of HyperX is complementary to our contributions as improved meta-exploration would help us better learn robust meta-policies.

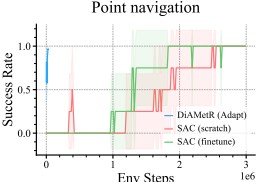

Figure 16: Both SAC trained from scratch (SAC scratch) and SAC pre-trained on training task distribution (SAC finetune) take more than a million timesteps to solve test tasks. In comparison, DiAMetR takes $30k$ timesteps to select the right meta-policy which then solves new tasks from test distribution in $k - 1$ environment episodes (i.e. 60 timesteps given $k = 2$ and horizon $H = 60$).

**Can RL quickly solve test time tasks?** To test if RL can quickly solve test-time tasks, we train Soft Actor Critic (SAC) [15] on 5 tasks sampled from a particular test task distribution. To make the comparison fair, we include a baseline that pre-trains SAC on train-task distribution. Figure 16 shows results on `Point-navigation`. We see that both SAC trained from scratch and SAC pre-trained on training task distribution take more than a million timesteps to solve test tasks. In comparison, DiAMetR takes $30k$ timesteps to select the right meta-policy which then solves new tasks from test distribution in $k - 1$ environment episodes (i.e. 60 timesteps given $k = 2$ and horizon $H = 60$). This shows that meta-RL formulation is required for quick-adaptation to test tasks.

## I Learning meta-policies with different support

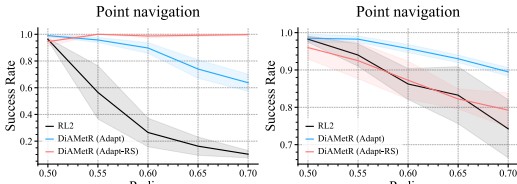

Figure 17: We investigate if learning meta-policies with different support (DiAMetR(Adapt-RS)) is better than learning meta-policies with overlapping support (DiAMetR(Adapt)). We evaluate it on two different families of shifted task distributions. While the left figure shows evaluations on shifted task distributions with different support (i.e. $\mathcal{U}(0, 0.5), \mathcal{U}(0.5, 0.55), \mathcal{U}(0.55, 0.6), \mathcal{U}(0.6, 0.65), \mathcal{U}(0.65, 0.7)$), the right figure shows evaluations on shifted task distributions with overlapping support (i.e. $\mathcal{U}(0, 0.55), \mathcal{U}(0.0, 0.6), \mathcal{U}(0.0, 0.65), \mathcal{U}(0.0, 0.7)$). While DiAMetR(Adapt-RS) performs the best on shifted task distributions with different support, DiAMetR(Adapt) performs the best on shifted task distributions with overlapping support. Hence, whether meta-policies should have overlapping support depends on the nature of shifted test task distributions.

We can alternatively try to learn multiple meta-policies, each with small and different (but slightly overlapping) support. In this way, there won't be any conservativeness tradeoff. To analyze this further, we focus on point navigation domain (with train target distance distribution as $\Delta \sim \mathcal{U}(0, 0.5)$) and experiment with out-of-support test task distribution.

Let's say we are learning the $i^{\text{th}}$ meta-policy (corresponding to $\epsilon_i$). Let $q_\phi^{i-1}(z) = \mathcal{N}(\mu^{i-1}, \sigma^{i-1})$ be the learned adversarial task distribution for $(i - 1)^{\text{th}}$ meta-policy (corresponding to $\epsilon_{i-1}$). To learn meta-policies with small and different support, we make 2 changes to Algorithm 1:

1. In step 7, we do a rejection sampling $z \sim q_\phi(z)$ with the condition that $\log q_\phi^{i-1}(z) \leq \beta$ (where $\beta$ is a hyperparameter. We found $\beta = -45$ to work well).

2. In step 8 and 9, we add another constraint that $D_{\text{KL}}(p_{\text{train}}(z)||q_\phi(z)) \geq \epsilon_{i-1}$ (in addition to $D_{\text{KL}}(p_{\text{train}}(z)||q_\phi(z)) \leq \epsilon_i$). This leads to learning of two weighting factors $\lambda_1$ and $\lambda_2$ (instead of just $\lambda$) that tries to ensure $D_{\text{KL}}(p_{\text{train}}(z)||q_\phi(z)) \in (\epsilon_{i-1}, \epsilon_i)$. We call this modified algorithm as DiAMetR (Adapt-RS) (where RS comes from rejection sampling).

The performance of this variant depends heavily on the form of the test task distribution. We first test it on task target distance distributions $\mathcal{U}(0, 0.5), \mathcal{U}(0.5, 0.55), \mathcal{U}(0.55, 0.6), \mathcal{U}(0.6, 0.65), \mathcal{U}(0.65, 0.7)$ (essentially testing on *rings* of disjoint support around the training distribution). We see that

DiAMetR(Adapt-RS) maintains a consistent success rate of $\sim 1$ across various target task distributions and outperforms DiAMetR(Adapt) and RL$^2$. This is because each meta-policy has overall smaller (hence are less conservative) and mostly different support and for this type of test distribution, this scheme can be very effective.

However, when we test it on task target distance distributions $\mathcal{U}(0, 0.5), \mathcal{U}(0, 0.55), \mathcal{U}(0.0, 0.6), \mathcal{U}(0.0, 0.65), \mathcal{U}(0.0, 0.7)$ (essentially testing on *discs* which mostly include the training distribution), we see that DiAMetR(Adapt-RS) performs same as RL$^2$ and mostly relies on the base RL$^2$ (i.e. $\epsilon = 0$) for its performance.

Whether meta-policies should have overlapping support will depend on the nature of shifted test task distributions. If supports of test task distributions overlap, then it's better to have meta-policies with overlapping support. Otherwise, it's more efficient to have meta-policies with different support.

## J    Meta-policy Selection with CEM during Meta-test

We explore using Cross-entropy method (CEM) [4] for meta-policy selection during meta-test phase, as an alternative to Thompson's sampling. Algorithm 4 details the use of the CEM algorithm for meta-policy selection. For this evaluation, we use `point navigation` environment where tasks vary in reward functions and test task distribution is *out-of-support* of training task distribution. Table 2 provides detailed information about these train and test task distributions. Figure 18 shows that CEM has similar performance as Thompson's sampling.

---

**Algorithm 4 DiAMetR**: Meta-test phase with CEM

1: Given: $p_{\text{test}}(\mathcal{T}), \Pi = \{\pi_{\text{meta},\theta}^{\epsilon_i}\}_{i=1}^{M}$
2: Sample $\pi \sim \Pi$ (with uniform probability) to collect 25 meta-episodes
3: Calculate $(\mu_\epsilon, \sigma_\epsilon)$ using 10 (of 25) (i.e. top 40%) meta-episodes with highest returns
4: **for** iter $t = 2, 3, ..., 10$ **do**
5:     **for** meta-episode $n = 1, 2, .., 25$ **do**
6:         Sample $\epsilon \sim \mathcal{N}(\mu_\epsilon, \sigma_\epsilon)$
7:         Choose $\epsilon_i$ closest to $\epsilon$
8:         Run $\pi_{\text{meta},\theta}^{\epsilon_i}$ for meta-episode
9:     **end for**
10:    Calculate $(\mu_\epsilon, \sigma_\epsilon)$ using 10 (of 25) (i.e. top 40%) meta-episodes with highest returns
11: **end for**

---

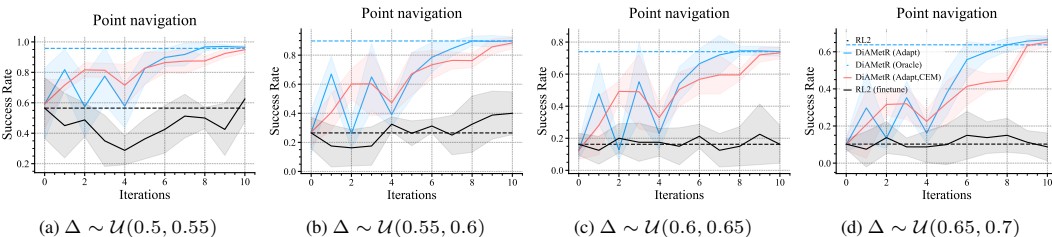

(a) $\Delta \sim \mathcal{U}(0.5, 0.55)$      (b) $\Delta \sim \mathcal{U}(0.55, 0.6)$      (c) $\Delta \sim \mathcal{U}(0.6, 0.65)$      (d) $\Delta \sim \mathcal{U}(0.65, 0.7)$

Figure 18: We compare use of Thompson's sampling (DiAMetR(Adapt)) and Cross-entropy method (DiAMetR(Adapt, CEM)) for test time adaptation of DiAMetR on point robot navigation for various test task distributions. While they have similar performance, they are both better than test-time finetuning of RL$^2$. We run the adaptation procedure for 10 iterations collecting 25 rollouts per iteration.

## K    Proof of Proposition 4.1

In this section, we prove the proposition in the main text about the excess regret of an $\epsilon_2$-robust policy under and $\epsilon_1$-perturbation (restated below).

**Proposition 4.1.** *Let* $\overline{\epsilon_i} = \min\{\epsilon_i + \beta, 1 - \frac{|\mathcal{S}_0|}{|\mathcal{S}|}\}$. *There exists* $q(\mathcal{T})$ *satisfying* $D_{TV}(p_{train}, q) \leq \epsilon_1$ *where an* $\epsilon_2$-*robust meta policy incurs excess regret over the optimal* $\epsilon_1$-*robust meta-policy:*

$$\mathbb{E}_{q(\mathcal{T})}[\text{Regret}(\pi_{meta}^{\epsilon_2}, \mathcal{T}) - \text{Regret}(\pi_{meta}^{\epsilon_1}, \mathcal{T})] \geq \left(c(\epsilon_1, \epsilon_2) + \frac{1}{c(\epsilon_1, \epsilon_2)} - 2\right) \quad (5)$$

$$\sqrt{\overline{\epsilon_1}(1 - \overline{\epsilon_1})|\mathcal{S}_0|(|\mathcal{S}| - \mathcal{S}_0|)} \quad (6)$$

*The scale of regret depends on* $c(\epsilon_1, \epsilon_2) = \sqrt{\frac{\overline{\epsilon_2}^{-1} - 1}{\overline{\epsilon_1}^{-1} - 1}}$, *a measure of the mismatch between* $\epsilon_1$ *and* $\epsilon_2$.

**Summary of proof:** The proof proceeds in three stages: 1) deriving a form for the optimal meta-policy for a *fixed* task distribution 2) proving that the optimal $\epsilon$-robust meta-policy takes form:

$$\pi_{\text{meta}}^{\epsilon}(s) \propto \begin{cases} \sqrt{\frac{1-\overline{\epsilon}}{|\mathcal{S}_0|}} & s \in \mathcal{S}_0 \\ \sqrt{\frac{\overline{\epsilon}}{|\mathcal{S}| - |\mathcal{S}_0|}} & s \notin \mathcal{S}_0 \end{cases}$$

and finally 3) showing that under the task distribution $p(\mathcal{T}_g) = (1 - \overline{\epsilon_1})\text{Uniform}(\mathcal{S}_0) + \overline{\epsilon_1}\text{Uniform}(\mathcal{S}\backslash\mathcal{S}_0)$, the gap in regret takes the form in the proposition.

*Proof.* For convenience, denote $\mathcal{S}_1 = \mathcal{S}\backslash\mathcal{S}_0$ and $\text{Regret}(\pi_{\text{meta}}, q(\mathcal{T})) = \mathbb{E}_q[\text{Regret}(\pi, \mathcal{T})]$. Furthermore, since the performance of a meta-policy depends only on its final-timestep visitation distribution (and any such distribution is attainable), we directly refer to $\pi(g)$ as the visited goal distribution of the meta-policy $\pi_{\text{meta}}$. Recall that the regret of $\pi_{\text{meta}}$ on task $\mathcal{T}_g$ is given by $\frac{1}{\pi(g)}$.

We begin with the following lemma that demonstrates the optimal policy for a fixed task distribution.

**Lemma K.1.** *The optimal meta-policy* $\pi_q^* = \arg\min_\pi \text{Regret}(\pi, q(\mathcal{T}_g))$ *for a given task distribution* $q(\mathcal{T}_g)$ *is given by*

$$\pi_q^*(g) = \frac{1}{\int \sqrt{q(g')}\,dg'}\sqrt{q(g)} \quad (16)$$

The proof of the lemma is similar to the argument in Gupta et al. [13], Lee et al. [20]:

$$\pi_q^* = \arg\min_{\pi(g)} \text{Regret}(\pi, q(\mathcal{T}_g)) = \arg\min_{\pi(g)} \mathbb{E}_{\mathcal{T}_g \sim q}\left[\frac{1}{\pi(g)}\right] \quad (17)$$

Letting $Z = \int_g \sqrt{q(g)}$, we can rewrite the optimization problem as minimizing an $f$-divergence (with $f(x) = \frac{1}{x}$)

$$= \int \frac{1}{\pi(g)}q(g)\,dg \quad (18)$$

$$= Z^2 \int \frac{\sqrt{q(g)}/Z}{\pi(g)}\sqrt{q(g)}/Z\,dg \quad (19)$$

$$= Z^2 D_f(\pi\|\sqrt{q}(g)/Z) \quad (20)$$

This is minimized when both are equal, i.e. when $\pi_q^*(g) = \sqrt{q}/Z$, concluding the proof.

**Lemma K.2.** *The optimal* $\epsilon$-*robust meta-policy* $\pi^{*\epsilon} = \arg\min \mathcal{R}(\pi, p_{train}, \epsilon)$ *takes form*

$$\pi^\epsilon(g) \propto \begin{cases} \sqrt{\frac{1-\overline{\epsilon}}{|\mathcal{S}_0|}} & g \in \mathcal{S}_0 \\ \sqrt{\frac{\overline{\epsilon}}{|\mathcal{S}| - |\mathcal{S}_0|}} & g \notin \mathcal{S}_0 \end{cases}$$

Define the distribution $q^\epsilon(\mathcal{T}_g) = (1 - \overline{\epsilon})\text{Uniform}(\mathcal{S}_0) + \overline{\epsilon}\text{Uniform}(\mathcal{S}\backslash\mathcal{S}_0)$, which is an $\epsilon$-perturbation of $p_{\text{train}}$ under the TV metric. We note that there are two main cases: 1) if $\overline{\epsilon} = 1 - \frac{|\mathcal{S}_0|}{|\mathcal{S}|}$, then $q^\epsilon$ is uniform over the entire state space, and otherwise 2) it corresponds to uniformly taking $\epsilon$-mass from

$\mathcal{S}_0$ and uniformly distributing it across $\mathcal{S}_1$. Using the lemma, we can derive the optimal policy for $q^\epsilon$, which we denote $\pi^\epsilon$:

$$\pi^\epsilon = \arg\min_{\pi(g)} \text{Regret}(\pi, q^\epsilon(\mathcal{T})) = \frac{1}{\int \sqrt{q^\epsilon(g')}\,dg'} \sqrt{q(g)}, \tag{21}$$

Writing $Z_\epsilon = \int \sqrt{q(g')}\,dg'$, we can write this explicitly as

$$= \frac{1}{Z_\epsilon} \begin{cases} \sqrt{\frac{1-\bar{\epsilon}}{|\mathcal{S}_0|}} & g \in \mathcal{S}_0 \\ \sqrt{\frac{\bar{\epsilon}}{|\mathcal{S}|-|\mathcal{S}_0|}} & g \notin \mathcal{S}_0 \end{cases} \tag{22}$$

We now show that there exists no other distribution $q'(\mathcal{T})$ with $TV(p_{\text{train}}, q') \leq \epsilon$ for which $\text{Regret}(\pi^\epsilon, q') \geq \text{Regret}(\pi^\epsilon, q^\epsilon)$. We break this into the two cases for $q^\epsilon$: if $q^\epsilon$ is uniform over all goals, then $\pi^\epsilon$ visits all goals equally often, and so incurs the same regret on every task distribution. The more interesting case is the second: consider any other task distribution $q'(g)$, and let $q_0', q_1'$ be the probabilities of sampling goals in $\mathcal{S}_0$ and $\mathcal{S}_1$ respectively under $q'$: $q_0' = \mathbb{E}_{g \sim q'}[\mathbb{1}(g \in \mathcal{S}_0)]$ and $q_1' = 1 - q_0'$. The regret of $\pi^\epsilon$ on $q'$ is given by

$$\text{Regret}(\pi^\epsilon, q'(\mathcal{T})) = \mathbb{E}_{g \sim q'}\left[\frac{1}{\frac{1}{Z_\epsilon}\sqrt{q(g)}}\right] \tag{23}$$

$$= Z_\epsilon\left(q_0'\sqrt{\frac{|\mathcal{S}_0|}{1-\bar{\epsilon}}} + q_1'\sqrt{\frac{|\mathcal{S}|-|\mathcal{S}_0|}{\bar{\epsilon}}}\right) \tag{24}$$

By construction of $\bar{\epsilon}$, we have that $\sqrt{\frac{|\mathcal{S}|-|\mathcal{S}_0|}{\bar{\epsilon}}} \geq \sqrt{\frac{|\mathcal{S}_0|}{1-\bar{\epsilon}}}$, and so this expression is maximized for the largest value of $q_1'$. Under a $\epsilon$-perturbation in the TV metric, the maximal value of $q_1$ is given by $\beta + \epsilon = \bar{\epsilon}$:

$$\leq Z_\epsilon\left((1-\bar{\epsilon})\sqrt{\frac{|\mathcal{S}_0|}{1-\bar{\epsilon}}} + \bar{\epsilon}\sqrt{\frac{|\mathcal{S}|-|\mathcal{S}_0|}{\bar{\epsilon}}}\right) \tag{25}$$

This is exactly the regret under our chosen task proposal distribution $q^\epsilon(\mathcal{T})$ (which has $q_1 = \bar{\epsilon}$)

$$= \text{Regret}(\pi^\epsilon, q^\epsilon(\mathcal{T})). \tag{26}$$

These two steps can be combined to demonstrate that $\pi^\epsilon$ is a solution to the robust objective. Specifically, we have that

$$\mathcal{R}(\pi_\epsilon, p_{\text{train}}, \epsilon) = \max_{q':TV(p_{\text{train}}, q')\leq\epsilon} \text{Regret}(\pi_\epsilon, q') = \text{Regret}(\pi_\epsilon, q^\epsilon) \tag{27}$$

so, for any other meta-policy $\pi_{\text{meta}}$, we have

$$\mathcal{R}(\pi, p_{\text{train}}, \epsilon) = \max_{q':TV(p_{\text{train}}, q')\leq\epsilon} \text{Regret}(\pi, q') \geq \text{Regret}(\pi, q^\epsilon) \geq \text{Regret}(\pi^\epsilon, q^\epsilon) = \mathcal{R}(\pi^\epsilon, p_{\text{train}}, \epsilon) \tag{28}$$

This concludes the proof of the lemma. $\square$

Finally, to complete the proof of the original proposition, we write down (and simplify) the gap in regret between $\pi^{\epsilon_1}$ and $\pi^{\epsilon_2}$ for the task distribution $q^{\epsilon_1}$ (as described above). We begin by writing down the regret of $\pi^{\epsilon_1}$:

$$\text{Regret}(\pi^{\epsilon_1}, q^{\epsilon_1}(\mathcal{T})) = Z_{\epsilon_1}\left((1-\bar{\epsilon_1})\sqrt{\frac{|\mathcal{S}_0|}{1-\bar{\epsilon_1}}} + \bar{\epsilon_1}\sqrt{\frac{|\mathcal{S}_1|}{\bar{\epsilon_1}}}\right) \tag{29}$$

$$= (\sqrt{|\mathcal{S}_0|(1-\bar{\epsilon_1})} + \sqrt{|\mathcal{S}_1|(\bar{\epsilon_1})})\left((1-\bar{\epsilon_1})\sqrt{\frac{|\mathcal{S}_0|}{1-\bar{\epsilon_1}}} + \bar{\epsilon_1}\sqrt{\frac{|\mathcal{S}_1|}{\bar{\epsilon_1}}}\right) \tag{30}$$

$$= (1-\bar{\epsilon_1})|\mathcal{S}_0| + \bar{\epsilon_1}|\mathcal{S}_1| + 2\sqrt{\bar{\epsilon_1}(1-\bar{\epsilon_1})|\mathcal{S}_0||\mathcal{S}_1|} \tag{31}$$

Next, we write the regret of $\pi^{\epsilon_2}$

$$\text{Regret}(\pi^{\epsilon_2}, q^{\epsilon_1}(\mathcal{T})) = Z_{\epsilon_2}((1 - \overline{\epsilon_1})\sqrt{\frac{|\mathcal{S}_0|}{1 - \overline{\epsilon_2}}} + \overline{\epsilon_1}\sqrt{\frac{|\mathcal{S}_1|}{\overline{\epsilon_2}}}) \tag{32}$$

$$= (\sqrt{|\mathcal{S}_0|(1 - \overline{\epsilon_2})} + \sqrt{|\mathcal{S}_1|(\overline{\epsilon_2})})((1 - \overline{\epsilon_1})\sqrt{\frac{|\mathcal{S}_0|}{1 - \overline{\epsilon_2}}} + \overline{\epsilon_1}\sqrt{\frac{|\mathcal{S}_1|}{\overline{\epsilon_2}}}) \tag{33}$$

$$= (1 - \overline{\epsilon_1})|\mathcal{S}_0| + \overline{\epsilon_1}|\mathcal{S}_1| + \overline{\epsilon_1}\sqrt{|\mathcal{S}_0||\mathcal{S}_1|\frac{(1 - \overline{\epsilon_2})}{\overline{\epsilon_2}}} + (1 - \overline{\epsilon_1})\sqrt{|\mathcal{S}_0||\mathcal{S}_1|\frac{(\overline{\epsilon_2})}{1 - \overline{\epsilon_2}}} \tag{34}$$

$$= (1 - \overline{\epsilon_1})|\mathcal{S}_0| + \overline{\epsilon_1}|\mathcal{S}_1| + \overline{\epsilon_1}\sqrt{|\mathcal{S}_0||\mathcal{S}_1|\frac{(1 - \overline{\epsilon_2})}{\overline{\epsilon_2}}} + (1 - \overline{\epsilon_1})\sqrt{|\mathcal{S}_0||\mathcal{S}_1|\frac{(\overline{\epsilon_2})}{1 - \overline{\epsilon_2}}} \tag{35}$$

$$= (1 - \overline{\epsilon_1})|\mathcal{S}_0| + \overline{\epsilon_1}|\mathcal{S}_1| + \sqrt{|\mathcal{S}_0|\mathcal{S}_1|\overline{\epsilon_1}(1 - \overline{\epsilon_1})}\left(\sqrt{\frac{\overline{\epsilon_1}}{(1 - \overline{\epsilon_1})}\frac{(1 - \overline{\epsilon_2})}{\overline{\epsilon_2}}} + \sqrt{\frac{(1 - \overline{\epsilon_1})}{\overline{\epsilon_1}}\frac{\overline{\epsilon_2}}{(1 - \overline{\epsilon_2})}}\right) \tag{36}$$

Now writing $c(\epsilon_1, \epsilon_2) = \sqrt{\frac{\overline{\epsilon_2}^{-1} - 1}{\overline{\epsilon_1}^{-1} - 1}} = \sqrt{\frac{1 - \overline{\epsilon_2}}{\overline{\epsilon_2}}\frac{\overline{\epsilon_1}}{1 - \overline{\epsilon_1}}}$

$$= (1 - \overline{\epsilon_1})|\mathcal{S}_0| + \overline{\epsilon_1}|\mathcal{S}_1| + \sqrt{|\mathcal{S}_0|\mathcal{S}_1|\overline{\epsilon_1}(1 - \overline{\epsilon_1})}\left(c(\epsilon_1, \epsilon_2) + \frac{1}{c(\epsilon_1, \epsilon_2)}\right) \tag{37}$$

$$= \text{Regret}(\pi^{\epsilon_1}, q^{\epsilon_1}(\mathcal{T})) + \sqrt{|\mathcal{S}_0|\mathcal{S}_1|\overline{\epsilon_1}(1 - \overline{\epsilon_1})}\left(c(\epsilon_1, \epsilon_2) + \frac{1}{c(\epsilon_1, \epsilon_2)} - 2\right) \tag{38}$$

This concludes the proof of the proposition.

## L Hyperparameters Used

Table 3 describes the hyperparameters used for the structured VAE for learning reward function distribution

| | |
|---|---|
| latent $z$ dimension (for reward and dynamics distribution) | 16 |
| $p(z)$ (for reward and dynamics distribution) | $\mathcal{N}(0, I)$ |
| $q_\psi(z|\overline{h})$ (for reward distribution) | MLP(hidden-layers=[256, 256, 256]) |
| $r_\omega^h(z)$ | MLP(hidden-layers=[256, 256, 256]) |
| $q_\psi(z|(s_t, a_t)_{t=1}^T)$ (for dynamics distribution) | GRU(hidden-layers=[256, 256, 256]) |
| $p_\omega^{\text{dyn}}(s, a)$ | MLP(hidden-layers=[256, 256, 256]) |
| Train trajectories (from train task replay buffer) | $1e6$/(Episodic Horizon $H$) |
| Train Epochs | 100 |
| initial $\log \sigma$ | $-5$ |
| $\{\epsilon_i\}_{i=1}^M$ (*out-of-support* test task distributions) | $\{0.0, 0.1, 0.2, 0.3, 0.4, 0.5, 0.6, 0.7, 0.8\}$ |
| $\{\epsilon_i\}_{i=1}^M$ (*in-support* test task distributions) | $\{0.0, 0.05, 0.1, 0.15, 0.2, 0.25, 0.3, 0.35, 0.4\}$ |
| $n_{\text{tr}}$ (num tasks in empirical $p_{\text{train}}(\mathcal{T})$) | 200 |

Table 3: Hyperparameters for structured VAE

We use off-policy RL$^2$ [31] as our base meta-learning algorithm. We borrow the implementation from https://github.com/twni2016/pomdp-baselines. We use the hyperparameters from the config file https://github.com/twni2016/pomdp-baselines/blob/main/configs/meta/ant_dir/rnn.yml but found 1500 `num-iters` was sufficient for convergence of the meta-RL algorithm. Furthermore, we use 200 `num-updates-per-iter`. Our codebase can be found at https://drive.google.com/drive/folders/1KTjst_n0PlR0O7Ez3-WVj0jbgnl1ELD3?usp=sharing.

We parameterize $q_\phi(z)$ as a normal distribution $\mathcal{N}(\mu, \sigma)$ with $\phi = (\mu, \sigma)$ as parameters. We use REINFORCE with trust region constraints (i.e. Proximal Policy Optimization [37]) for optimizing $q_\phi(z)$. We borrow our PPO implementation from the package https://github.com/ikostrikov/pytorch-a2c-ppo-acktr-gail and default hyperparameters from https://github.com/ikostrikov/pytorch-a2c-ppo-acktr-gail/blob/master/a2c_ppo_acktr/arguments.py. Table 4 describes the hyperparameters for PPO that we changed.

| | |
|---|---|
| num-processes | 1 |
| ppo-epoch | 10 |
| num-iters | 3 |
| num-env-trajectories-per-iter | 100 |

Table 4: Hyperparameters for PPO for training $q_\phi$ per every meta-RL iteration

We use off-policy VariBAD [6] implementation from the package https://github.com/twni2016/pomdp-baselines/tree/main/BOReL with their default hyperparameters. We use HyperX [50] implementation from the package https://github.com/lmzintgraf/hyperx with their default hyperparameters. To make the comparisons fair, we ensure that the policy and the Q-function in VariBAD and HyperX have same architecture as that in off-policy RL$^2$ [31].

## M  Visualizing meta-policies chosen by Thompson's sampling

In this section, we visualize the behavior of different meta-policies (robust to varying levels of distribution shift) towards end of their training. We additionally plot the meta-policies chosen by Thompson's sampling during meta-test phase for different task distribution shifts. We choose `Ant navigation` task for this evaluation with training task target distance distribution as $\mathcal{U}(0, 0.5)$ and test task distributions being *out-of-support* of training task distribution (see Table 2 for detailed description of these distributions).

To visualize the behavior of different meta-policies, we extract the $x - y$ position of Ant from the states visited by these meta-policies in their last million environment steps (out of their total $15.2$ million environment steps). We pass these $x - y$ positions through a gaussian kernel and generate visitation heatmaps of the Ant's $x - y$ position, as shown in Figure 20. Figure 19 plots the $\epsilon$ values corresponding to meta-policies chosen by Thompson's sampling during meta-test for different test task distributions.

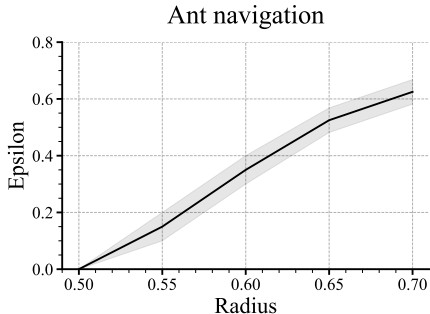

Figure 19: $\epsilon$ values corresponding to meta-policies chosen by Thompson's sampling during meta-test for different test task distributions of `Ant navigation`. The first point $r_{\text{train}}$ on the horizontal axis indicates the training target distance $\Delta$ distribution $\mathcal{U}(0, r_{\text{train}})$ and the subsequent points $r_{\text{test}}^i$ indicate the shifted test target distance $\Delta$ distribution $\mathcal{U}(r_{\text{test}}^{i-1}, r_{\text{test}}^i)$.

## N  Cost of learning multiple meta-policies during meta-train phase

Training a population of meta-policies mainly requires more memory (both RAM and GPU) as the meta-policies are trained in parallel. It is true that granularity of epsilon affects the performance of

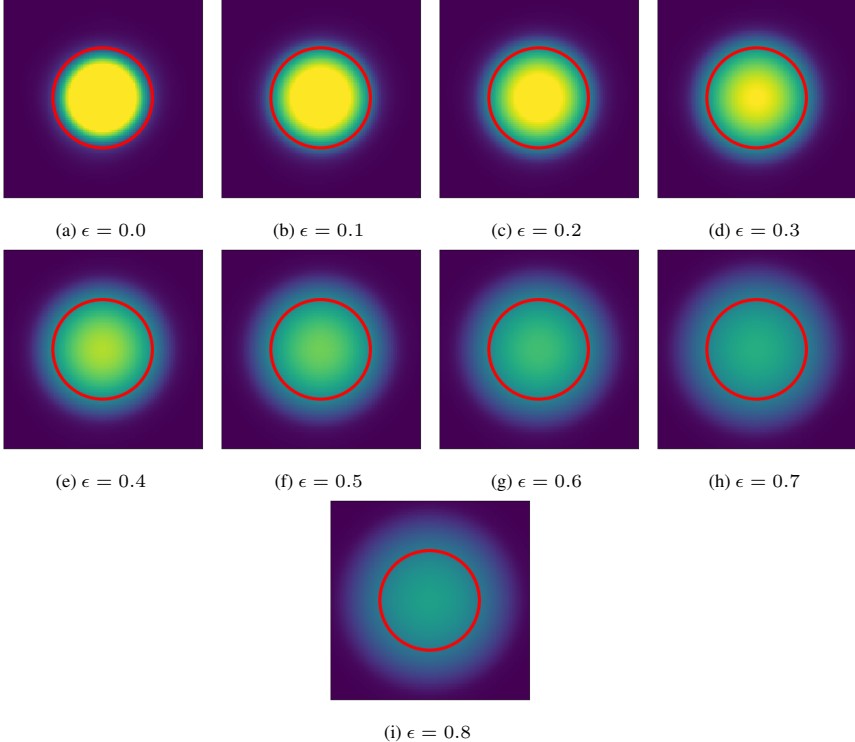

(a) $\epsilon = 0.0$    (b) $\epsilon = 0.1$    (c) $\epsilon = 0.2$    (d) $\epsilon = 0.3$

(e) $\epsilon = 0.4$    (f) $\epsilon = 0.5$    (g) $\epsilon = 0.6$    (h) $\epsilon = 0.7$

(i) $\epsilon = 0.8$

Figure 20: We visualize the Ant's $x - y$ position visitation heatmaps for different meta-policies towards the end of their training (i.e., in their last 1 million environment steps, out of 15.2 million environment steps of training). Here, $\epsilon$ indicates the level of robustness of the meta-policy $\pi_\theta^\epsilon$. The red circle visualizes the training task distribution for `Ant navigation`.

DiAMeTR (as argued in section 4.3). If we increase the number of trained meta-policies, even though DiAMetR's final performance on various shifted test task distribution would improve, it would take more time/samples for test-time adaptation. While choosing the number of trained meta-policies, we need to balance between final asymptotic performance and time/samples taken for test-time adaptation. The main benefit of learning a distribution of such meta-policies is that it amortizes over many different shifted test time distributions and the meta-policies do not need to be relearned for each of these.

## Author Contributions

**Anurag Ajay** helped in the technical formulation of distributionally adaptive meta RL, implemented the DiaMetR algorithm, ran experiments and played the primary role in paper writing.

**Abhishek Gupta** conceived the framework of distributionally adaptive meta RL, ran preliminary proof-of-concept experiments, helped with paper writing and played an advisory role to Anurag.

**Dibya Ghosh** analyzed the framework of distributionally adaptive meta RL (Section 4.3), wrote all the related proofs, participated in research discussions and helped with paper writing.

**Sergey Levine** provided feedback on the work and paper writing, and participated in research discussions.

**Pulkit Agrawal** was involved in research discussions, influenced the choice of experimental domains, provided feedback on writing, positioning of the work and overall advising.