# OpenReview forum: "Distributionally Adaptive Meta Reinforcement Learning"
_NeurIPS.cc/2022/Conference — NeurIPS 2022 Accept_

### Official Review · Reviewer_V7F9 · 2022-07-10

**Rating:** 6
**Confidence:** 4
**Soundness:** 3 good
**Presentation:** 3 good
**Contribution:** 2 fair

**Summary:**

In meta RL, the most common assumption is that the training task distribution is the same as the test task distribution, which may not hold in practice. To address the unknown distribution shift in practice, this paper proposes an adaptive method, by learning a set of policies with varying levels of distribution shift. Under a certain level of distribution shift, the policy is learned by leveraging the framework of distributional robustness.  At test time, the selection of policies is formulated as a MAB problem.

**Questions:**

1. I was wondering why the authors did not capture the distribution over reward and transition functions, in their method to parameter the task distribution. It seems not very difficult.
2. Assuming p_train(z) = N(0, I) does not seem reasonable to me. N(0, I) is prior. Even training the ELBO  only on training task distribution, the posterior is usually different from the prior.
3. Is it possible to approve some uni-modality property of this special MAB problem? If so, you can use a more efficient method with the unimodal bandit.

**Limitations:**

1. The method can be extended to capture both variations in reward function and transition function.
2. The theoretical results are limited to a toy environment.

**Strengths And Weaknesses:**

Strengths
1. The distribution shift problem is very common in practice.
2. The idea of learning a set of policies with varying levels of distribution shift, when the test distribution cannot be observed in advance, is quite reasonable.
3. The paper is well-organized.

Weaknesses:
1. The theoretical analysis is only limited to a toy environment.
2. The proposed method only parameters the task distribution by parameterizing distribution over the reward function. However, different tasks usually differ in both reward and transition functions.
3. The test-time meta policies selection is based on MAB. I was wondering if it is possible to improve this MAB method, perhaps by considering some uni-modality properties of this special MAB problem(Is it possible to approve that?).

---

> ### Author Response · Authors · 2022-08-02
> **Author's response**
>
> We thank the reviewer for their constructive feedback. We will now address the specific points below.
>
> >The proposed method only parameters the task distribution by parameterizing distribution over the reward function. However, different tasks usually differ in both reward and transition functions.
>
> We have added experiments showing that DiAMetR can be applied to problems with changing dynamics as well. Please see  **Dynamics Variability** in the common response for an overview of our results, and Section 6 / Appendix F in the revised paper for details.
>
> >The test-time meta policies selection is based on MAB. I was wondering if it is possible to improve this MAB method, perhaps by considering some uni-modality properties of this special MAB problem(Is it possible to approve that?).
>
> To test this hypothesis, we ran an experiment using cross entropy method to select the epsilon, rather than a MAB. CEM operates on a continuous space and relies on a unimodal gaussian assumption as suggested by the reviewer, as it uses a unimodal gaussian policy parameterization. To adapt CEM to our scenario we still perform optimization in continuous space of epsilons as CEM would, but while evaluating each sample of epsilon, round this to the nearest discrete choice of epsilon. These results are shown in Fig 21, Appendix K. In the experiments conducted, there was not a significant difference in performance but it is possible that this may be more pronounced in more scaled up task distributions.
>
> > Assuming p_train(z) = N(0, I) does not seem reasonable to me. N(0, I) is prior. Even training the ELBO only on training task distribution, the posterior is usually different from the prior.
>
> We follow the standard training convention in variational inference methods like the variational autoencoder [kingma_2013], where the variational posterior is regularized against a unit gaussian prior and then samples are drawn from this unit gaussian prior and pushed forward through the decoder to generate samples. The important part here is that because the posterior is regularized against the prior, samples from the prior when pushed forward through the decoder can capture rich, multi-modal data distributions. [higgins_2017] provides a more in-depth justification of this design choice in methods like the variational auto-encoder.
>
> >The theoretical analysis is only limited to a toy environment.
>
> The analysis in the paper is meant to be pedagogically illustrative of the behavior of the proposed algorithm rather than a full-fledged theoretical analysis of meta-learning and distribution shift. We definitely agree this is very interesting but perhaps is out of scope of this paper and would merit a thorough investigation in a follow up paper with a stronger theoretical focus.
>
> **References:**
>
> [kingma_2013]: Diederik P Kingma, Max Welling.'Auto-Encoding Variational Bayes', arXiv 2013.
>
> [higgins_2017]: Irina Higgins, Loic Matthey, Arka Pal, Christopher Burgess, Xavier Glorot, Matthew Botvinick, Shakir Mohamed, Alexander Lerchner. 'beta-VAE: Learning Basic Visual Concepts with a Constrained Variational Framework', ICLR 2017.

---

> > ### Comment · Reviewer_V7F9 · 2022-08-08
> > **Reply**
> >
> > Post Rebuttal: Thanks for the authors' detailed response. Overall, this paper proposes a novel method for a novel problem setting.  I'm maintaining my score and increasing my confidence.

---

> ### Author Response · Authors · 2022-08-06
> **Looking forward to further discussions!**
>
> Dear Reviewer,
>
> Thank you for your time and effort in reviewing our work. We have provided detailed clarification and additional experiments to address the issues raised in your comments. If our response has addressed your concerns, we would be grateful if you could re-evaluate our work.
>
> If you have any additional questions or comments, we would be happy to have further discussions.
>
> Thanks,
>
> The authors

---

### Official Review · Reviewer_qqiy · 2022-07-10

**Rating:** 7
**Confidence:** 5
**Soundness:** 4 excellent
**Presentation:** 4 excellent
**Contribution:** 3 good

**Summary:**

This paper identifies and addresses the problem of the distributional shift in test task distributions for meta-RL agents, which often struggle under such circumstances. For this matter, the work leverages the distributional robustness framework to alleviate this effect, by training a population of meta-policies in different levels of distribution shift. In order to do so, the proposed method (DiAMetR) firstly learns a single meta-policy in the train distribution and uses the collected trajectories to learn a probabilistic latent space to parameterize task distributions. Then, each subsequent meta-policy is learned in an adversarial game between itself and a task proposal distribution with some degree of shift. At test-time, a bandit algorithm based on Thompson sampling selects the appropriated meta-policy (level of distribution shift) to interact with the test task. The proposed method is evaluated in 4 different continuous control environments and presents better adaptation performance in the scenarios of test distribution shift when compared to strong baselines. The work also presents ablations to validate the components of the method and a theoretical proposition to better understand a meta-policy performance under distribution shift.

**Questions:**

The main suggestions or questions/concerns to be clarified are pointed out in the Strengths/Weaknesses and Limitations sections.

**Limitations:**

First of all, I appreciate that the work raised the limitation regarding handling only tasks with reward variation (as opposed to dynamics variation). It would be interesting to discuss ways to address this limitation in future work, in light of the proposed framework.

Besides this limitation, there is also another point worth discussing:

The distributional shift phenomenon can happen in many different ways. In other words, considering the training distribution, there might have several different test distributions with the same level of shift (given a divergence metric). It is hard to know if a single “adversarial game” per shift level would be able to cover enough possibilities of test task distributions.

For the presented experimental setup, this does not seem to be a problem, as the shift would happen in only one dimension (reward) that varies across very few variables (e.g, radius and angle of the goal). However, in more complex cases, this can be concerning, and adding new adversarial games incurs new meta-policies, which also increases adaptation complexity (which is already a concern as pointed out in the weaknesses). Curious to know your thoughts on this point.

Figure 7d illustrates this concern: it is possible to see that the imagined test distributions are more “skewed” in the west-south directions, but the real test distribution could be the opposite, for example.


**Strengths And Weaknesses:**

Strengths:

- Addressing distributional shift for meta-RL agents is a very important topic, especially since OOD generalization is a property that the community is actively looking for understanding and improvements;
- The paper is very organized and well written, all ideas are clear and the work is appropriately placed in prior literature;
- The method is novel, technically very sound, and brings a solid contribution to the field of meta-RL towards the mitigation of distributional shift in test task distribution;
- The theoretical framework is clearly stated and the derived proposition is relevant; the experimental setups support the main claims and back up the proposed method. As a strong positive point, there is a helpful, extensive ablation and the raised questions during the experimental setup are very relevant to validating the method.

Weaknesses/Concerns:

- Using a multi-armed bandit for meta-policy selection seems to be very costly (in terms of test-time sample complexity), and the complexity grows with the number of meta-policies. For instance, the example from Section 6.1 shows that  DiAMetR requires 250 meta-episodes (or 500 episodes) to complete adaptation. Prior literature shows evidence for few-episode adaptation, sometimes online adaptation [1, 2]. I understand that these methods are not focused on distributional shift, but in scenarios where test-time sample complexity is also a requirement, this performance seems concerning.
- In the same line as the previous point, it would be interesting to show the results from variBAD and hyperX in the scenario of test time adaptation (Figure 5 and Appendix G). It would be helpful to understand how these other methods would handle this trade-off between performance and adaptation complexity for the distributional shift scenario.
- As noted by the paper, the proposed implementation is limited by meta-RL environments comprising only variations in the reward distribution, missing those ones with variations in the dynamics. This is an important concern, as there is no way to validate if the proposed method is still valid for more complex benchmarks. For instance, it would be interesting to see how it performs in MetaWorld ML10/ML45 [3].

Despite these weaknesses, the understanding is that the method and empirical validation present satisfactory quality for acceptance, given the proposed scope.

Minor issues:

- Figure 2 is not cited and actually does not seem to have any support in the text besides Algorithm 1.
- A diagram with all the modules and training/test stages would be very helpful to improve clarity and presentation. For example, something like Figure 2 in the [VariBAD paper](https://arxiv.org/pdf/1910.08348.pdf).
- In Equation 1, $J(\pi_{t})$ is not defined (Is the minus sign a typo?)
- Algorithm 1, Line 3 cites Equation 7, but this actually refers to the meta-policy selection, is this right?
- In Appendix F, the summary of proof (L575): the paper describes a 3-stage proof, but stage “1)” is repeated twice.

References:

[1] Zintgraf et. al. VariBAD: A Very Good Method for Bayes-Adaptive Deep RL via Meta-Learning. ICLR, 2020.

[2] LC Melo. Transformers are Meta-Reinforcement Learners. ICML, 2022.

[3] Yu et. al. Meta-World: A Benchmark and Evaluation for Multi-Task and Meta Reinforcement Learning. CoRL, 2019.

---

> ### Author Response · Authors · 2022-08-02
> **Author's response**
>
> We thank the reviewer for their constructive feedback. We will now address the specific points below.
>
> >Using a multi-armed bandit for meta-policy selection seems to be very costly (in terms of test-time sample complexity), and the complexity grows with the number of meta-policies. For instance, the example from Section 6.1 shows that DiAMetR requires 250 meta-episodes (or 500 episodes) to complete adaptation. Prior literature shows evidence for few-episode adaptation, sometimes online adaptation [1, 2]. I understand that these methods are not focused on distributional shift, but in scenarios where test-time sample complexity is also a requirement, this performance seems concerning.
>
> We would like to clarify what we meant by few-episode/quick adaptation during test time. During test time, we get a shifted test task distribution from which new tasks can be sampled, not just an individual task. We require 250 meta episodes to identity and adapt to the shifted test task distribution. However, once we have adapted (i.e. figured out meta-policy with appropriate level of robustness) to the shifted test task distribution, we can learn new tasks (from the shifted test task distribution) within k (i.e. 2) episodes as is common across meta-reinforcement learning tasks. Given that the scenario we consider is more challenging than standard meta-RL tasks, the additional meta-episodes needed to identity the shifted task distribution can be amortized over learning many new tasks from this distribution quickly (in only 2 episodes) once this has been identified.
>
> >As noted by the paper, the proposed implementation is limited by meta-RL environments comprising only variations in the reward distribution, missing those ones with variations in the dynamics.
>
> We have added experiments showing that DiAMetR can be applied to problems with changing dynamics as well. Please see  **Dynamics Variability** in the common response for an overview or our results, and Appendix K in the revised paper for details.
>
> >In the same line as the previous point, it would be interesting to show the results from variBAD and hyperX in the scenario of test time adaptation (Figure 5 and Appendix G). It would be helpful to understand how these other methods would handle this trade-off between performance and adaptation complexity for the distributional shift scenario.
>
> We indeed fine-tuned VariBAD and HyperX with same number of samples (i.e. 10 iteration with 25 meta-episode per iteration = 250 meta episode) during test-time adaptation. Similar to RL^2, their performance remains largely unchanged after test-time finetuning. We have included the finetuning curves of VariBAD and HyperX in the updated draft (Figure 5 and Appendix G).
>
> >Figure 2 is not cited and actually does not seem to have any support in the text besides Algorithm 1.
>
> We have cited Figure 2 in the introduction of the updated draft so that the reviewer can see how it is related to the text.
>
> >A diagram with all the modules and training/test stages would be very helpful to improve clarity and presentation. For example, something like Figure 2 in the VariBAD paper.
>
> This has been added to the paper in Appendix J, Fig 17.
>
> >In Equation 1, J(πt) is not defined (Is the minus sign a typo?)
>
> We have updated Equation 1 to make it more clear. $J(\pi_{\mathcal{T}}^*)$ is the expected return given by the optimal policy for task $\mathcal{T}$. The minus sign is not a typo since we are trying to minimize regret which, in turn, means maximizing expected return of the meta-policy over $k$ episodes.
>
> >Algorithm 1, Line 3 cites Equation 7, but this actually refers to the meta-policy selection, is this right?
>
> Yes, the reviewer is right. We have updated Line 3 to refer to Equation 6.
>
> >In Appendix F, the summary of proof (L575): the paper describes a 3-stage proof, but stage “1)” is repeated twice.
>
> We have fixed the typo to correctly refer to the second stage with "2)".
>
> > However, in more complex cases, this can be concerning, and adding new adversarial games incurs new meta-policies, which also increases adaptation complexity
>
> This is a very valid concern, as the dimensions of the problem increase, the space of possible "adversarial" distributions also increases and the curse of dimensionality rapidly becomes a problem. While many distributional robustness methods would suffer from this pitfall as the reviewer points out, perhaps the most practical way to deal with this issue would be to leverage some weak supervision to identify the important axes (or features) along which distribution shift may occur (for instance as illustrated in this work [Lee_2020]), and only measure divergence of the proposed task distributions along these axes. This would be very interesting to explore more deeply in future work!
>
> **References:**
>
> [Lee_2020] Lisa Lee, Ben Eysenbach, Russ R Salakhutdinov, Shixiang Shane Gu, Chelsea Finn. "Weakly-supervised reinforcement learning for controllable behavior", NeurIPS 2020.

---

> > ### Comment · Reviewer_qqiy · 2022-08-05
> > **Feedback on Author's Response**
> >
> > Thank you to all the authors for the very detailed responses and improvements in the paper.
> >
> > I carefully read the author's clarifications, as well as all the additional experiments in the paper. My main concerns are addressed and I believe the empirical support is now excellent, improving the soundness of the work. In reflection, I also increased this score in the post-rebuttal evaluation.
> >
> > The authors also recognized the limitation related to high dimensionality and proposed an interesting idea for future work, which satisfies my expectations for acceptance since I also believe this is out of scope for the current submission.

---

### Official Review · Reviewer_Z27V · 2022-07-11

**Rating:** 5
**Confidence:** 3
**Soundness:** 2 fair
**Presentation:** 3 good
**Contribution:** 2 fair

**Summary:**

The paper proposes a meta-RL algorithm for dealing with test-time distribution shift.
This is achieved by learning a population of policies that have different levels of distributional robustness by an adversarial algorithm.
At test time, a policy is chosen amongst the population by an efficient bandit algorithm.
The paper demonstrates empirically that this way of adapting to test-time distribution achieves better performance than competing methods.
Theoretical result provides a bound on the difference of expected regret of a mis-specified robustness epsilon.


**Questions:**

# Major Concerns

* Equation 1 is expected regret, shouldn’t the regret be written as  - expected return of the meta-policy and the actual term in the expectation at the moment is just an MC estimate, this would seem more natural to me, because the outer expectation is taken over different tasks, not environment dynamics, so the expected return of the meta-policy doesn’t even appear in the minimization problem.

* Fix fonts in figure 5, they are way too small. Also, RL2 and RL2 finetune legend doesn’t match the dashed lines. It’s not clear to me what is diameter oracle here? If it’s the best policy that can be chosen from the set of trained meta-policies, why is then the diameter adapt mean higher than the oracle’s mean? This figure needs better clarification.

* It seems to me that your results hinge on the fact that if you are lucky enough that the test distribution is close to the train distribution, you should be less conservative, otherwise you should be more conservative and therefore sacrifice performance. Doesn’t it make more sense to have policies that are less conservative (train on task distributions of smaller support, slightly overlapping support) and then choose amongst them? In this way there would be no ‘conservativeness tradeoff’. This is not shown in figure 7.

*  Ideally I would also like to see the distribution over policies from the population of meta-policies that get picked for solving tasks. Is it skewed towards more conservative policies? The plot should show that conservativeness grows with epsilon, currently you only have the result that picking the most conserative policy isn’t the best thing. (figure 8).

* l.141 I don’t understand why equation 3 incentivizes exploration as you say it? I understand how it leads to distributional robustness, since you want to minimize the maximum regret, which is the difference to the optimal expected return. But how does this encourage exploration? I don’t see any reason why this should be the case…

* How many adaptation policies are used in the experiments? Experiments need better clarification in general.

# Minor Concerns

* Section 6.2, by test reward distributions you mean q(z | h)? Please put inline math somewhere in this section to make the clear where the samples come from? What projection method was used to generate these heatmaps in 2d? Or is z actually 2-dimensional?

* Figure 8: fix the fonts, same as figure 5. I don’t understand how is diameter mid uncertainty set chosen here exactly, so it’s hard for me to infer anything from this result, what does it imply?

* Equation 2, why is q parametrized here? It should be just a set of distributions that are close in terms of f-divergence to the training distribution p, there is no reason why they should be parameterized.

Requirements to increase my score:

* Better explanation of the experiments section (what do the plots mean).
* Better explanation of section 6.2 (projection method, where do the samples come from).
* Argue clearly why does it make sense to have more and less conservative policies in comparison to less conservative ones with different task-space specialisations.
* Answer all the listed points




**Limitations:**

The authors should make it clear how much it costs them to learn this population of policies and how it relates to the test-time distribution shift. I imagine that the number of policies is also going to affect the performance in the end, because it's directly related to the granularity of epsilon (which controls conservativeness).

**Strengths And Weaknesses:**

# Strengths

* The method proposed in the paper is novel (as far as I know) and seems to show improvement in comparison to other meta RL methods.
* The presentation of the method is clear, I didn't have many problems following the storyline

# Weaknesses

* Missing details in the experiment section.
* Simplified task spaces where goals are be sampled from uniform distribution and limitation to only shift in reward function as result, not transition dynamics for example.
* It is unclear to me based on the presentation how conservative the policies that are chosen by the bandit are in the end.* “the distributional meta-robustness objective can be modelled as an adversarial game between the policy and the task distribution”. Unfortunately this is the only sentence that hints at a game-theoretic treatment of the topic, but it doesn’t exist in the paper. I like that using an adversarial formulation for inferring the task sampling distribution q allows for a lot of flexibility regarding task distributions, the environments though used in the experiments to validate the method all deal with relatively simple distribution shifts concerning the reward (just uniform distributions of goals), so perhaps the tasks used for evaluation are too simple?

---

> ### Author Response · Authors · 2022-08-02
> **Author's response (1/4)**
>
> We thank the reviewer for their constructive feedback. We will now address the specific points below.
>
> >Simplified task spaces where goals are be sampled from uniform distribution and limitation to only shift in reward function as result, not transition dynamics for example.
>
> We have added experiments showing that DiAMetR can be applied to problems with changing dynamics as well. Please see  **Dynamics Variability** in the common response for an overview of our results, and Section 6 (and Appendix F) in the revised paper for details.
>
> >It is unclear to me based on the presentation how conservative the policies that are chosen by the bandit are in the end.
> >Ideally I would also like to see the distribution over policies from the population of meta-policies that get picked for solving tasks. Is it skewed towards more conservative policies? The plot should show that conservativeness grows with epsilon, currently you only have the result that picking the most conserative policy isn’t the best thing. (figure 8).
>
> We provide an analysis in Figure 23, Appendix O. The general trend we notice is that the more conservative policies get selected as the distribution shift between train task distribution and test task distribution increases.
>
> >“the distributional meta-robustness objective can be modelled as an adversarial game between the policy and the task distribution”. Unfortunately this is the only sentence that hints at a game-theoretic treatment of the topic, but it doesn’t exist in the paper. I like that using an adversarial formulation for inferring the task sampling distribution q allows for a lot of flexibility regarding task distributions, the environments though used in the experiments to validate the method all deal with relatively simple distribution shifts concerning the reward (just uniform distributions of goals), so perhaps the tasks used for evaluation are too simple?
>
> To address this concern we have added a number of new experiments - shifts in transition dynamics, dense reward functions and "in-support" distribution shift as we describe in the common response. While we believe a full game theoretic consideration of the proposed algorithm is beyond the scope of this work, we are very interested in exploring this in future work!
>
> >Equation 1 is expected regret, shouldn’t the regret be written as - expected return of the meta-policy and the actual term in the expectation at the moment is just an MC estimate, this would seem more natural to me, because the outer expectation is taken over different tasks, not environment dynamics, so the expected return of the meta-policy doesn’t even appear in the minimization problem.
>
> The expected regret is the regret taken in expectation over the distribution of MDPs (or tasks), which can vary in either dynamics or reward. It is just saying that for every task you take the difference between the return of the best possible policy on that task and the average return of the policy being updated with a meta-learning algorithm $\pi_\text{meta}$ over several learning iterations. So for every task, the Regret measures how fast the \emph{same} shared meta-policy learns and this is taken in average across many different tasks drawn from the task distribution.
>
> >It’s not clear to me what is diameter oracle here? If it’s the best policy that can be chosen from the set of trained meta-policies, why is then the diameter adapt mean higher than the oracle’s mean? This figure needs better clarification.
>
> DiAMetR oracle is the best meta-policy that can be chosen from the set of trained meta-policies. While some plots have DiAMetR adapt mean higher than DiAMetR oracle mean, the differences are within the standard error and therefore not significant.

---

> > ### Author Response · Authors · 2022-08-02
> > **Author's response (2/4)**
> >
> > >It seems to me that your results hinge on the fact that if you are lucky enough that the test distribution is close to the train distribution, you should be less conservative, otherwise you should be more conservative and therefore sacrifice performance. Doesn’t it make more sense to have policies that are less conservative (train on task distributions of smaller support, slightly overlapping support) and then choose amongst them? In this way there would be no ‘conservativeness tradeoff’. This is not shown in figure 7.
> >
> > The reviewer makes a good suggestion to learn multiple meta-policies, each with small and different (but slightly overlapping) support to avoid a conservativeness tradeoff. To analyze this further, we focus on point navigation domain (with train target distance distribution as $\mathcal{U}(0, 0.5)$) and experiment with out-of-support test task distribution (as was done in the paper submission as well).
> >
> > While the paper submission aims to learn a set of meta-policies which are robust to uncertainty sets with $D_{KL}(p || q) \leq \epsilon_i$ for different choices of $\epsilon_i$, we can instead learn a set of meta-policies which are robust to uncertainty sets with $D_{KL}(p || q) \leq \epsilon_i^h$ and $D_{KL}(p || q) \geq \epsilon_i^l$. In this way it avoids double counting and each of the task distributions has different (but slightly overlapping support) as suggested by the reviewer. As we will dicsuss next, in certain scenarios this strategy can work well but in others it can have suboptimal performance.
> >
> > **Method** : At a high level, we implement this by changing the constraint in Equation 3 to a pair of constraints upper and lower bounding the task distribution divergence (rather than just an upper bound), while solving the robust optimization problem. This ensures task distributions have minimally overlapping support. Pleas refer to Appendix N for details about this procedure. We call this modified algorithm as DiaMetR (Adapt-RS).
> >
> > **Result** : The performance of this variant depends heavily on the form of the test task distribution. We first test it on task target distance distributions $\mathcal{U}(0, 0.5), \mathcal{U}(0.5, 0.55), \mathcal{U}(0.55, 0.6), \mathcal{U}(0.6, 0.65), \mathcal{U}(0.65, 0.7)$ (essentially testing on **rings** of disjoint support around the training distribution). We see that DiAMetr(Adapt-RS) maintains a consistent success rate of ~1 across various target task distributions and outperforms DiAMetR (Adapt) and RL^2. This is because each meta-policy has overall smaller (hence are less conservative) and mostly different support and for this type of test distribution, this scheme can be very effective.
> >
> > However, when we test it on task target distance distributions $\mathcal{U}(0, 0.5), \mathcal{U}(0, 0.55), \mathcal{U}(0.0, 0.6), \mathcal{U}(0.0, 0.65), \mathcal{U}(0.0, 0.7)$ (essentially testing on **discs** which mostly include the training distribution), we see that DiAMetr(Adapt-RS) performs same as standard meta-RL and mostly chooses to use the base meta-RL algorithm (i.e. $\epsilon=0$) for its performance.
> >
> > Whether meta-policies should have overlapping support will depend on the nature of shifted test task distributions. If supports of test task distributions overlap, then it's better to have meta-policies with overlapping support. Otherwise, it's more efficient to have meta-policies with different support. This discussion was added to Appendix J.
> >
> > >I don’t understand why equation 3 incentivizes exploration as you say it? I understand how it leads to distributional robustness, since you want to minimize the maximum regret, which is the difference to the optimal expected return. But how does this encourage exploration? I don’t see any reason why this should be the case…
> >
> > In equation 3, we are simultaneously learning an adversarial task distribution $q_\phi(\mathcal{T})$ and a meta-policy. The adversarial task distribution imagines tasks that make the performance of meta-policy worse (while being constrained within $epsilon$ distance of the training task distribution). In case where the reward changes, the imagined tasks puts high reward on target locations outside train target location region (i.e. outside the circular region of radius $r_{\text{train}}$). To adapt to the proposed adversarial task distribution (which may be outside the training distribution), the meta policy is forced to explore outside the region of train task locations. As $\epsilon$ increases, $q_\phi(\mathcal{T})$ begins to propose tasks that are further outside the training region and hence the meta policy is forced to explore more.

---

> > > ### Author Response · Authors · 2022-08-02
> > > **Author's response (3/4)**
> > >
> > > >How many adaptation policies are used in the experiments?
> > >
> > > We used 9 different meta-policies (corresponding to 9 different $\epsilon$) that we choose from during test-time adaptation. For experiments with out-of-support distributionally shifted test task distributions, we choose the set of $\epsilon$ to be $\{0, 0.1, 0.2, 0.3, 0.4, 0.5, 0.6, 0.7, 0.8\}$ for all environments. For experiments with in-support distributionally shifted test task distributions, we choose the set of $\epsilon$ to be $\{0, 0.05, 0.1, 0.15, 0.2, 0.25, 0.3, 0.35, 0.4\}$ for all environment. We have updated the original draft to include these details (see Appendix M, table 3).
> > >
> > > >Equation 2, why is q parametrized here? It should be just a set of distributions that are close in terms of f-divergence to the training distribution p, there is no reason why they should be parameterized.
> > >
> > > It is not clear how to characterize and optimize for the set of distributions that are "close" to the training distribution p, without defining a family of distributions and a divergence metric to measure closeness. To be able to sample from the distribution q, we represent them as gaussians in latent space, pushed forward through a latent space decoder to generate a more expressive distribution. This allows us to measure the f-divergence in closed form which allows us to solve Equation 2 using standard optimization methods and sample from the distribution q by sampling in the latent space and pushing samples forward through the decoder. This is challenging without an explicit parameterization.
> > >
> > > In the new experiment we introduced on "in-support" distribution shift (Section 6/Appendix F), the family of q is simply parameterized as reweighting of the training distribution but it is still a distribution that allows us to measure f-divergence in closed form (Section 5).
> > >
> > > >I don’t understand how is diameter mid uncertainty set chosen here exactly, so it’s hard for me to infer anything from this result, what does it imply?
> > >
> > > We chose the mid uncertainty set as the one corresponding to $\epsilon=0.4$ for experiments on out-of-support test task distribution ($\epsilon=0.2$ for experiments on in-support test task distribution). It is a baseline where the uncertainty set is neither too big nor too small. This experiment shows that it is hard for a single meta-policy (with a fixed uncertainty set) to perform well across different distributionally shifted test task distributions, and that adaptivity actually matters for test-time performance.
> > >
> > > >Section 6.2, by test reward distributions you mean q(z | h)? Please put inline math somewhere in this section to make the clear where the samples come from? What projection method was used to generate these heatmaps in 2d? Or is z actually 2-dimensional?
> > >
> > > By imagined test reward distributions, we mean the learned $q_\phi(z)$ as a result of solving equation (3) (i.e. resulting from the adversarial task distribution update in Algorithm 1). $z$ is 16 dimensional. To generate an imagined reward function, we sample $z \sim q_\phi(z)$ and pass the $z$ into $r_\omega(s,a,z) = r_\omega(s,z)$ (given the learned reward is only dependent on state as mentioned in Appendix C). We then take the agent (for instance the ant) and reset its (x,y) location to different points in the (discretized) grid [-1,1]^2 and calculate $r_\omega(s,z)$ at all those points. This gives us a reward map for a single imagined reward function. We sample $10000$ of these reward functions and plot them together. We see that more goal locations (yellow dots) are sampled outside the training goal region (shown as a red circle) as we increase $\epsilon$. We included this discussion in Appendix N.
> > >
> > > >Fix fonts in figure 5, they are way too small. Also, RL2 and RL2 finetune legend doesn’t match the dashed lines.
> > >
> > > We have updated the font size in figures 4,5,6,7 (and other figures in appendix). We have also updated the RL^2 and RL^2 (fine-tune) legend.

---

> > > > ### Author Response · Authors · 2022-08-02
> > > > **Author's response (4/4)**
> > > >
> > > > >The authors should make it clear how much it costs them to learn this population of policies and how it relates to the test-time distribution shift. I imagine that the number of policies is also going to affect the performance in the end, because it's directly related to the granularity of epsilon (which controls conservativeness).
> > > >
> > > > Training a population of meta-policies mainly requires more memory (both RAM and GPU) as the meta-policies are trained in parallel. It is true that granularity of epsilon affects the performance of DiAMeTR (as argued in section 4.3). If we increase the number of trained meta-policies, even though DiAMetR's final performance on various shifted test task distribution would improve, it would take more time/samples for test-time adaptation. While choosing the number of trained meta-policies, we need to balance between final asymptotic performance and time/samples taken for test-time adaptation. The main benefit of learning a distribution of such meta-policies is that it amortizes over many different shifted test time distributions and the meta-policies do not need to be relearned for each of these. We added this point in Appendix P.
> > > >
> > > > > Missing details in the experiments section
> > > >
> > > > As suggested, we have added some additional details about the experiments - choice of epsilons (Appendix M, table 3), explanation of Figure 5 (Appendix N), number of adaptation policies (Appendix M, table 3).

---

> ### Author Response · Authors · 2022-08-06
> **Looking forward to further discussions!**
>
> Dear Reviewer,
>
> Thank you for your time and effort in reviewing our work. We have provided detailed clarification and additional experiments to address the issues raised in your comments. If our response has addressed your concerns, we would be grateful if you could re-evaluate our work.
>
> If you have any additional questions or comments, we would be happy to have further discussions.
>
> Thanks,
>
> The authors

---

> ### Author Response · Authors · 2022-08-08
> **Further clarifications?**
>
> Dear reviewer Z27V,
>
> Since we are coming to the end of the reviewer-author discussion period, we would like to know if you have remaining questions or concerns following our response. Please let us know. We would be happy to offer further clarification in the time remaining.
>
> Best,
>
> Authors

---

### Official Review · Reviewer_Asdp · 2022-07-12

**Rating:** 4
**Confidence:** 4
**Soundness:** 2 fair
**Presentation:** 3 good
**Contribution:** 3 good

**Summary:**

This paper proposes a new framework (called DiAMetR) for meta RL, aiming at improving the ability of meta policy in adapting to varying levels of task distribution shift. This is achieved by leveraging the distributional robust framework to re-formulate the meta RL objective to optimize the worst-case empirical risk under a uncertainty set of test-time task distribution. To be specific, the task distribution is parameterized through modeling the latent representation of the reduced reward functions. Then, the training alternates between 1) the training of meta policy against multiple imagined task distributions, each of which has a different distribution deviation constraint, and 2) the training of the imagined task distributions in an adversarial manner.

The performance of DiAMetR is evaluated in four goal-reaching environments, demonstrating the superiority in adaptation to varying levels of task distribution shift when compared with RL$^2$, VariBAD and HyperX. Ablation study and analysis are also provided.


**Questions:**

1) How do the authors think about the generalization ability of the reward function models trained with samples collected by changing policies under the original task distributional? (as my comment says in the part of Strengths and Weaknesses)
2) For Line 7 in Algorithm 1, i.e., training the meta policy with imagined task distributions, how are the state-action trajectories collected? I can imagine three options: 1) directly interact with environment since the transition dynamics is the same; 2) interact with a learned transition dynamics model; 3) reuse the experiences in the replay buffer.
3) How are the set of task distribution deviation (i.e., $\{\epsilon\}$) chosen for each environments in the experiments? I did not found the associated content (I also checked the appendix), did I miss it?
4) Do the baselines VariBAD and HyperX also use finetuning with the same number of samples during test-time adaptation?
5) Although the authors provide Figure 8, it would be useful and informative to provide the analysis between the arm (i.e., some $\epsilon$) adaptively chosen by Algorithm 2 and the oracle test distribution.

&nbsp;

Minors:
- The legends in Figure 5,6,8 can be enlarged for better clarity.

&nbsp;


I may improve my rating if my concerns and questions are well addressed.


**Limitations:**

The major limitation is the proposed method focuses on reward variations while assuming transition dynamics unchanged. This is considered as one of the future directions in Section 7.

&nbsp;


I think there are three more limitations which are not well discussed in this paper:
1) The first one is the choice of the uncertainty sets (i.e., the multiple $\epsilon$). Intuitively, these hyperparameters are significant to the learning performance of the proposed method, and one may need several attempts or prior knowledge to determine the task-dependent choices.
2) The second one is that the performance of current method in dense-reward environments, or non-goal reaching environments remains unclear. I also notice that the structured VAE is kind of specialized for sparse-reward tasks.
3) The third one is my concern on the sample cost of test-time adaptation (i.e., 250 meta episode). I am a little reluctant to agree with the ‘quick adaptation’, especially when considering other possible environments with horizons longer than e.g. 50, 200 (Table 2 in the Appendix) used in this work.


**Strengths And Weaknesses:**

Pros:
+ The paper is well written and the presentation of the algorithm is very clear.
+ The related work is almost sufficient.
+ The main idea is general and novel in meta RL. Insightful formal analysis is provided.
+ The proposed method is evaluated from multiple perspectives, from performance comparison to visual analysis.

&nbsp;

Cons:

- The proposed method is kind of limited.
  - This is mainly because DiAMetR relies on the assumption that  the target tasks only differ at reward function while the transition dynamics remain unchanged. Personally, I think this cripples the significance of the proposed method a lot.
  - In addition, it is unclear how DiAMetR will perform in dense-reward environments or non-goal reaching environments. It seems that the specific model structure (i.e., structured VAE in appendix is specialized to sparse-reward environments).
- The latent representation and generative model of reward function (Eq.6) implicitly depends on the distribution of state-action pairs. In practice, the distribution of state-action pairs is generated by the policies optimized during the learning process under the original training task distribution.
  - I have no question on the worst-case objective (Eq.3 and Eq.4) when oracle imagined task distribution and task generation are considered. However, since the latent representation and generative model of reward function are trained with samples collected by changing policies under the original task distributional, I worry about the ability of the generalization of such reward function models, in turn the effectiveness and rationality, especially in more possible environments.
- Important experimental details are missing (see my questions below).

&nbsp;

There are some related works which can be included for more complete empirical evaluation and related work survey:

- MetaCURE: Meta Reinforcement Learning with Empowerment-Driven Exploration. ICML 2021
- Towards Effective Context for Meta-Reinforcement Learning: an Approach based on Contrastive Learning. AAAI 2021

---

> ### Author Response · Authors · 2022-08-02
> **Author's response (1/2)**
>
> We thank the reviewer for their constructive feedback. We have added several new experiments to address reviewer concerns as described in the common response. We will now address the specific points below.
>
> >DiAMetR relies on the assumption that the target tasks only differ at reward function while the transition dynamics remain unchanged. Personally, I think this cripples the significance of the proposed method a lot.
>
> We have added experiments showing that DiAMetR can be applied to problems with changing dynamics as well. Please see  **Dynamics Variability** in the common response for an overview or our results, and Appendix K in the revised paper for details.
>
> >It is unclear how DiAMetR will perform in dense-reward environments or non-goal reaching environments.
>
> We have included an experiment on dense reward tasks (Appendix M, Fig 22) and on tasks with changing dynamics (Appendix K, Fig 19). We find that DiAMetR also outperforms other meta-RL approaches on these problems, although we find the gap in performance with dense rewards to be less extreme than with sparse rewards.
>
> >How do the authors think about the generalization ability of the reward function models trained with samples collected by changing policies under the original task distributional?
>
> As the reviewer pointed out, reward models have no reason to magically generalize if just trained naively. We consider two scenarios - one where the proposed tasks can consist of "out-of-support" tasks and the other where proposed tasks are all "in-support" but have different likelihoods across tasks.
>
> In the first case - that of out of support extrapolation, we found (in Appendix C, Figure 9) that reward models fail to generalize without additional structure or inductive bias. In this case, the additional structure provided through the structured form of the generative modeling allows for this kind of extrapolation. Without this structure, the problem is somewhat ill-posed, since out-of-support distributions break most assumptions that machine learning models make.
>
> In the second case - that of in support extrapolation, we added a new experiment in Appendix L, Fig [20, 21] as described in the common response, to show the effectiveness of DiAMetR in scenarios where distributions are different but support is shared. DiAMetR shows substantial gains in this scenario as well and in this case does not require the additional structure on the reward model since no out of support extrapolation is being performed.
>
> We hope these scenarios provide a better understanding of when generalization is necessary for the learned task distributions. These scenarios and the corresponding changes to DiAMetR are detailed in Appendix J.
>
> >How are the set of task distribution deviation (i.e., $\epsilon$) chosen for each environments in the experiments?
> >The first one is the choice of the uncertainty sets (i.e., the multiple ). Intuitively, these hyperparameters are significant to the learning performance of the proposed method, and one may need several attempts or prior knowledge to determine the task-dependent choices.
>
> While we searched for initial sets of epsilons for experiments on in-support and out-of-support distributionally shifted test task distributions, we didn't have to do any environment-specific tuning. All the other hyperparameters of our base off-policy meta-RL algorithm were taken from [ni_2022] and kept the same.  For experiments with out-of-support distributionally shifted test task distributions, we choose the set of $\epsilon$ to be $\{0, 0.1, 0.2, 0.3, 0.4, 0.5, 0.6, 0.7, 0.8\}$ for all environments. For experiments with in-support distributionally shifted test task distributions, we choose the set of $\epsilon$ to be $\{0, 0.05, 0.1, 0.15, 0.2, 0.25, 0.3, 0.35, 0.4\}$ for all environments. We have updated the original draft to include these details (see Appendix I, table 3).

---

> > ### Author Response · Authors · 2022-08-02
> > **Author's response (2/2)**
> >
> > >For Line 7 in Algorithm 1, i.e., training the meta policy with imagined task distributions, how are the state-action trajectories collected? I can imagine three options: 1) directly interact with environment since the transition dynamics is the same; 2) interact with a learned transition dynamics model; 3) reuse the experiences in the replay buffer.
> >
> > This depends on the scenario:
> > 1. **Experiments with out-of-support shifted test task distributions where only rewards vary across tasks:** We directly interact with environment since the transition dynamics is the same.
> > 2. **Experiments with out-of-support shifted test task distributions where only dynamics vary across tasks:** (NEW) We interact with a learned transition dynamics model.
> > 3. **Experiments with in-support shifted test task distributions where rewards vary across tasks:** (NEW) We directly interact with environment since the new tasks are still in-support of the empirical training task distribution and just have a different weighting (which just reweighs the loss term corresponding to that task).
> > We made this distinction explicit in Appendix J.2.
> >
> > >Do the baselines VariBAD and HyperX also use finetuning with the same number of samples during test-time adaptation?
> >
> > Yes, we also fine-tuned VariBAD and HyperX with same number of samples (i.e. 10 iteration with 25 meta-episode per iteration = 250 meta episode) during test-time adaptation. Similar to RL^2, their performance remains largely unchanged after test-time finetuning. We have included the finetuning curves of VariBAD and HyperX in the updated draft in Section 6.1, Fig 5 and Appendix G, Figures 12-16.
> >
> > >Although the authors provide Figure 8, it would be useful and informative to provide the analysis between the arm (i.e., some ) adaptively chosen by Algorithm 2 and the oracle test distribution.
> >
> > We provide an analysis in Figure 26, Appendix R. The general trend we notice is that the more conservative policies get selected as the distribution shift between train task distribution and test task distribution increases.
> >
> > >The third one is my concern on the sample cost of test-time adaptation (i.e., 250 meta episode). I am a little reluctant to agree with the ‘quick adaptation’, especially when considering other possible environments with horizons longer than e.g. 50, 200 (Table 2 in the Appendix) used in this work.
> >
> > We would like to clarify what we meant by quick adaptation during test time. During test time, we get a shifted test task distribution from which new tasks can be sampled. We indeed require 250 meta episodes to adapt to the shifted test task distribution. However, once we have adapted (i.e. figured out meta-policy with appropriate level of robustness) to the shifted test task distribution, we can learn new tasks (from the shifted test task distribution) within k (i.e. 2) episodes, with the cost of 250 meta-episodes amortized over many test tasks drawn from this distribution.
> >
> > > There are some related works which can be included for more complete empirical evaluation and related work survey: MetaCURE: Meta Reinforcement Learning with Empowerment-Driven Exploration. ICML 2021. Towards Effective Context for Meta-Reinforcement Learning: an Approach based on Contrastive Learning. AAAI 2021
> >
> > We have updated the paper draft to include these in the related work section and will run these as baselines in the final draft of the paper.
> >
> > >The legends in Figure 5,6,8 can be enlarged for better clarity.
> >
> > We have enlarged the legends in Figure 5,6,8.
> >
> > **References:**
> >
> > [ni_2022]: Tianwei Ni, Benjamin Eysenbach, Ruslan Salakhutdinov.'Recurrent Model-Free RL Can Be a Strong Baseline for Many POMDPs', ICML 2022.

---

> > > ### Author Response · Authors · 2022-08-08
> > > **Re: Further Feedback to Author Response**
> > >
> > > Dear Reviewer Asdp,
> > >
> > > We are messaging here again as we weren't sure about the visibility of our response to the post **"Further Feedback to Author Response"** (i.e. the reviewers weren't getting included as readers in that response).
> > >
> > > We thank you for your updated review and for engaging with us during the discussion. We have posted a response (titled **"Author's response to reviewer Asdp"** at top of the page) and updated the manuscript to alleviate your remaining concerns. Let us know if you have any other concerns.
> > >
> > > Thanks
> > >
> > > Authors

---

### Author Response · Authors · 2022-08-02
**Common response (1/2)**


We thank the reviewers for their thoughtful suggestions and comments. We have run several new experiments and added visualizations to help address the concerns brought up by the reviewers, which we summarize below. These changes can be found in the updated revision, highlighted in violet color, and described below.

1. **Dynamics Variability**: In Section 6 and Appendix F, we evaluated DiAMetR on problems with varying dynamics, demonstrating that our framework is not limited to handling only changes in the reward function
2. **In-support vs. Out-of-Support**: In Section 6 (figure 6) and Appendix F, we added experiments dealing with "in-support" distribution shifts, in addition to our original "out-of-support" distribution shift experiments.
3. **Dense Reward**: In Appendix G, we added experiments showing the ability to adapt to tasks with dense reward.
4.  **Non overlapping support**: In Appendix J, we added experiments suggested by Reviewer Z27V as an alternative to showing a different style of getting task distributions with non overlapping support as suggested by to mitigate conservativeness.
5. **Visualizations**: In Appendix O, we added visualizations that indicate what meta-policies are being selected, and the degree of conservativeness learned by the multi-armed bandit.

---

> ### Author Response · Authors · 2022-08-02
> **Common response (2/2)**
>
> We describe these in detail below:
>
> 1. **Dynamics Variability**: As suggested by all reviewers, the distribution shift may be in the dynamics function and not just the reward. As our framework of adaptive distributional robuystness is not limited to only shifting rewards, we have added new experiments (Section 6, Fig 4 and 7, Appendix F, Fig 11 and 12) extending DiAMetR to scenarios where dynamics are changing. We evaluate DiAMetR on the wind environment discussed in several prior work on meta-RL[dorman_2021, ni_2022] where the reward function is fixed but the dynamics function vary across tasks.To handle varying dynamics, we change the parameterized task distribution to model dynamics distributions rather than rewards (detailed in Section 5.1) and then use *imagined* new dynamics distributions close to the training dynamics distributions to train robust meta-policies. Our experiments in Section 6 and Appendix F show that DiAMetR outperforms existing meta-RL methods on these task distributions involving changing dynamics as well.
>
> 2. **In-support vs Out-of-Support**: To be more comprehensive in our analysis, we characterize task distribution shifts into 2 categories: ``in-support`` and ``out-of-support`` (the original submission only considered out of support distribution shifts). In Section 5, we present an updated version of DiAMetR that handles both categories. Reviewer [Asdp, Z27V] pointed out that learning a structured generative model seems challenging for task distribution generation. This is largely necessary when tasks are ``out-of-support" of the training distribution as there is a need for out of support extrapolation, which is challenging without baking in structure into the model architecture. However, this is not fundamental to our insights and the algorithm can be instantiated for problems without "out-of-support" extrapolation. To illustrate this, in Section 6 (Fig 6) and Appendix F (Fig 11, 12), we consider problems where the distribution is shifted but "in-support" (as is common in distributional robustness[zhou_2022]) rather than picking points that are completely out of support. We describe an alternate parameterization of adversarial task distribution where it is represented as a reweighted train task distribution $q_\phi(\mathcal{T}) \propto w_{\mathcal{T}} p_{\text{train}}(\mathcal{T})$. This allows for handling of "in-support" task distribution shifts and doesn't require learning of an additional task distribution model. While Section 5 describes how DiAMetR handles "in-support" task distribution shifts without learning any task distribution model, Section 6 (and Appendix F) contains experiments on "in-support" task distribution shifts and compares DiAMetR to existing meta RL algorithms.
>
> 3. **Dense reward**: The reviewers also brought up the applicability to non-sparse reward goal reaching environments. To illustrate this, we instantiate a dense reward version of the "ant" environment described in Appendix D and show in Appendix G, Fig 13 that the same insights hold for dense reward environments but with exploration being less challenging, as is expected.
>
> 4. **Non overlapping support**: We added an experiment that tries to learn task distributions and meta-policies that have minimal overlapping support and search between those for adaptation as suggested by reviewer Z27V. This is detailed in Appendix J. For a detailed description, refer to the response to reviewer Z27V.
>
> 5. **Visualizations**: We provide visualizations of chosen policies for various levels of distribution shift in Fig 23, Appendix O. We also provide an explanation of how we generated heatmaps for imagined test reward distributions (Section 6.2, Figure 5). The explanation is detailed in Appendix N. For a detailed response, please refer to the response to reviewer Z27V.
>
> For individual reviewer concerns, please see responses below each review.
>
> **References:**
>
> [dorman_2021]: Ron Dorfman, Idan Shenfeld, Aviv Tamar. 'Offline Meta Reinforcement Learning -- Identifiability Challenges and Effective Data Collection Strategies', NeurIPS 2021.
>
> [ni_2022]: Tianwei Ni, Benjamin Eysenbach, Ruslan Salakhutdinov.'Recurrent Model-Free RL Can Be a Strong Baseline for Many POMDPs', ICML 2022.
>
> [zhou_2022]: Xiao Zhou*, Yong Lin*, Renjie Pi*, Weizhong Zhang, Renzhe Xu, Peng Cui, Tong Zhang. 'Model Agnostic Sample Reweighting for Out-of-Distribution Learning', ICML 2022.

---

### Author Response · Authors · 2022-08-07
**Author's response to Reviewer Asdp**

We thank the reviewer for their response. We are glad that our response clarified some of the reviewer's concerns. We now address the remaining concerns.

>The main body of this paper may need substantial modifications to incorporate the additional content. I am kind of worried about that the revision goes too far from the submitted one.

We initially added changes in the form of new Appendix sections to address the reviewer's various concerns. We have uploaded a new draft incorporating the major changes into the main text itself. The new additions that are most substantial are:

1. the discussion of the algorithm generalizing to changing dynamics along with reward (Section 5.1)
2. experiments on in-support distribution shifts (figure 6, parts of section 6)
3. experiments showing the ability to adapt to distribution shifts in dynamics (figure 4 and 7, parts of section 6)

We have included this in the main text of the paper, within the 9 page limit. For the camera version of the paper which provides us with an extra page, we plan to elaborate the discussion on *in-support* distribution shifts and include Figure 9 (which would give additional clarity overview of DiAMetR's architecture) into the main paper as well.

Note that the major structure of the paper remains largely unchanged. We have not made fundamental modifications to the method, just some additional experiments and discussion showing the applicability of the method to more scenarios (thus, displaying broader applicability), and additional visualizations for better clarity.

We thank the reviewer (and the other reviewers as well) for their constructive suggestions and we feel that the changes that the reviewers suggestions significantly strengthen the story in the paper, while not changing the fundamental conclusions that are being made.

>A question for this part is, the authors say “adaptively choosing an uncertainty set during test time allows for better test time distribution adaptation when compared to selecting an uncertainty set beforehand or selecting a large uncertainty set” in Line 734-736, what is the meaning of “adaptively choosing an uncertainty set during test time” (this is different from the meta-test phase since the uncertainty set is fixed there, right?) and “selecting an uncertainty set beforehand” (is it the standard way described in the main body)? I realized that this may be a question I missed in my initial comments. Can the authors provide some more explanation (I found the first occurrence in Line 553-555 in the revision) and help me out of such confusion?

**Setup:** In meta-reinforcement learning, the test task distribution is same as train task distribution during meta-test. In our setup, there's a distributional shift between the train task and the test task distribution during meta-test. The level of distribution shift is unknown.

**Baselines:** (selecting an uncertainty set beforehand) One possible solution could be to prepare for a large distribution shift and thus *select a meta-policy with a large uncertainty set* (i.e. high epsilon). We call this baseline as DiAMetR (Conservative). Other possible solution could be to *select an uncertainty set beforehand* that is neither too big nor too small (i.e. a mid value of epsilon). We call this baseline as DiAMetR (Mid).

**Our approach:** (adaptively choosing an uncertainty set during test time) We infer the distribution shift and choose an uncertainty set based on experience during meta-test. Hence, at meta-test time, we both infer the level of distribution shift by doing inference on epsilon (as described in Section 5.2) and once we identify the right shift (i.e. epsilon), we do per task adaptation using the standard meta-RL RNN adaptation mechanism. We require 250 meta episodes to identity the level of distribution shift at meta test time. Once this is done, we can learn new tasks from this distribution within k episodes. This is what we mean by "adaptively choosing an uncertainty set during test time". Our results show that selecting an uncertainty set based on experience during meta-test performs better than choosing some uncertainty set beforehand without using experience at meta-test time (i.e. DiAMetR (Mid)) or choosing a large uncertainty set (i.e. DiAMetR (Conservative)).

>I am also happy to see the additional results with dense rewards (Appendix M). For a suggestion of future improvement, it will be better to see non goal-reaching or navigation task. Possible environment choices can be MuJoCo locomotion with varying dynamics or rewards, as these environments are also often adopted in meta-RL literature.

Thank you for the suggestion! We will include ant-dir and walker-random-param mujoco tasks in the final version of the paper.

---

### Meta-Review · Area_Chair_8uGf · 2022-08-26

**Recommendation:** Accept
**Confidence:** Less certain

**Metareview:**

The paper introduces a new framework for meta RL and validates it on navigation and goal reaching tasks. The authors addressed several of reviewers’ concerns in the rebuttal and significantly improved the quality of the paper. I think the paper is novel and interesting for the community. It would be interesting to see the performance of the proposed algorithm in more continuous tasks as suggested by reviewer Asdp.

**Award:**

No

---

### Decision · Program_Chairs · 2022-09-14

Accept